# REAL-CAPTURED PAIRED DATASET FOR NIGHTTIME FLARE REMOVAL

## ABSTRACT

Flare removal methods eliminate reflective and scattering flares within images and commonly adopt synthetic data for training. However, they fail to achieve robustness for real-world flare-corrupted images as the synthetic data remains gaps with real-world data. In this paper, we propose a real-captured paired dataset named FlareReal, which contains both real-captured image pairs and pure flare images. Compared with the existing flare removal dataset Flare7k++, our dataset is particularly effective for real-world scenarios as our data contains the faithful mapping between real flare-corrupted images and real flare-free images. Additionally, previous methods either lack sufficient receptive fields or achieve them with huge computational costs, which leads to flares being partly removed or hardly processing high-resolution images. Therefore, we propose a novel flare removal network named **M**utual re**C**eption f**LA**re **RE**moval **N**etwork (McLaren), which utilizes convolutions with diverse kernel sizes and fuses them from the perspective of both spatial and channel dimensions to achieve a sufficient receptive field. Furthermore, we employ a re-parameterization mechanism to avoid occupying excessive computational resources. We conduct extensive experiments to demonstrate the functions of our FlareReal dataset and our McLaren network.

## 1 INTRODUCTION

Lens flare is a common optical phenomenon when lights scatter or reflect within the sophisticated lens system. Due to the dirt on the lens and the multiple lens design, the scattering flare and the reflective flare commonly happen. The industry adopts anti-reflection coatings to suppress the reflective flare phenomenon, which is optimized for specific wavelengths and angles of light and leaves the scattering flare unaddressed. Therefore, how to remove the flare attracts plenty of attention.

Early days, as no available paired images for the flare removal task, He et al. (2010) and Qiao et al. (2021) adopt unsupervised method and achieve promising results, yet hardly handle severe flares, especially flares with strong streaks. As paired datasets Wu et al. (2021); Dai et al. (2022; 2023) proposed, the generalization of the flare removal method is widely expanded. By training image restoration methods Ronneberger et al. (2015); Chen et al. (2021); Zamir et al. (2021; 2022); Wang et al. (2022); Sharma & Tan (2021); Zhang et al. (2022; 2023) on such datasets, existing flare removal methods manage to handle plenty of flare patterns, whereas suffer from generating annoying artifacts during the flare removal process, especially when handling real scenarios. We argue that the synthetic data in the existing training set remains gaps with real-world images and the gap leads to the lack of robustness for real-world scenarios. Moreover, the existing image restoration methods either lack sufficient receptive field or achieve it with a huge computation burden, thereby cannot fully analyze flare patterns or process high-resolution flare-corrupted images.

In this paper, we propose a real-captured flare removal dataset named FlareReal and a novel flare removal network named **M**utual re**C**eption f**LA**re **RE**moval **N**etwork (McLaren). Specifically, the flare removal dataset contains 3,000 image pairs and 500 pure flare images for the training set and 50 image pairs for the validation set. The real-captured image pairs are collected for real-world scenarios, thereby we fully consider corner cases in the real-world applications. The shapes and numbers of light sources within collected flare-corrupted images are arbitrary and some images are shot under incorrect exposure settings (see Figure 1). Besides, we also collect 500 pure flare images in the darkroom to expand the training set. Compared with the Flare-R set in the Flare7k++ dataset

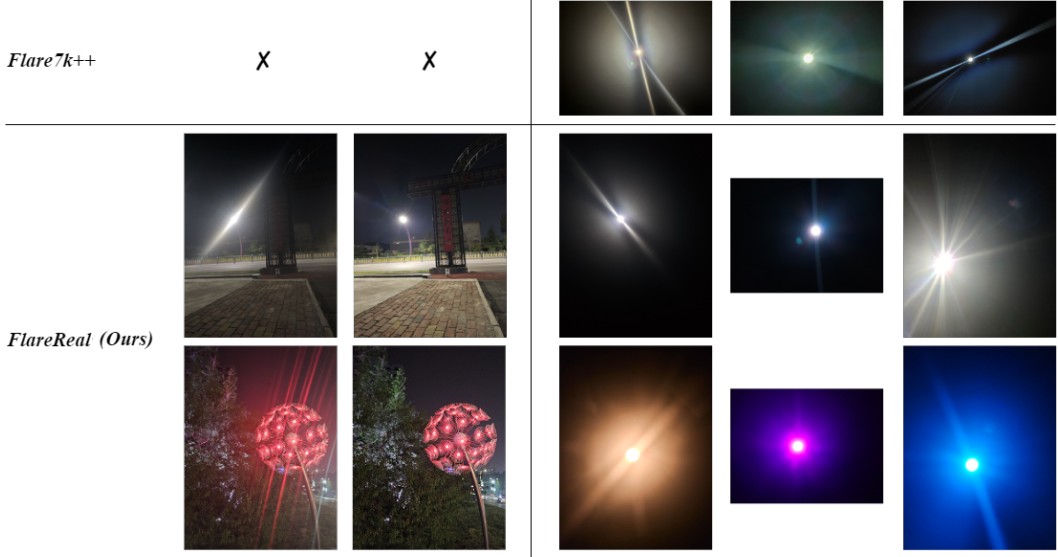

Figure 1: The difference between our FlareReal training set and Flare7k++ Dai et al. (2023) training set. Compared to the Flare7k++ dataset, our FlareReal dataset includes real-captured paired images with diverse exposure settings and light sources and flare images with more colors.

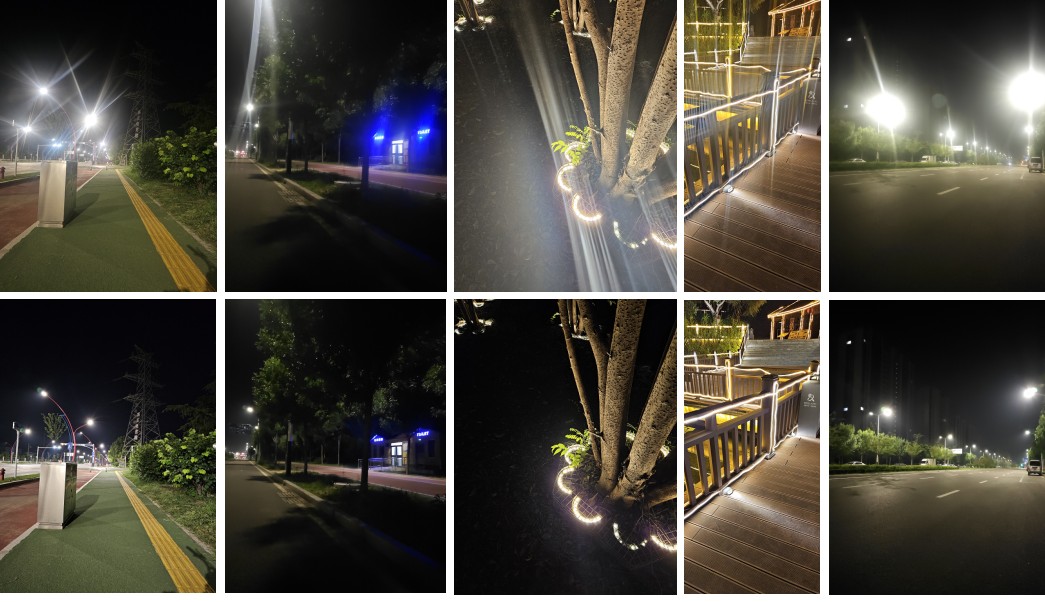

Figure 2: Samples of our FlareReal dataset. Images in the First row are flare-corrupted images and images in the second row are the corresponding flare-free images. The images in the third and the forth columns contain irregular light sources and the images in the fifth columns are captured in an underexposure situation.

Dai et al. (2023), our flares contain light sources with more colors (see Figure 1). Therefore, our dataset covers more practical applications than current datasets.

Furthermore, to obtain sufficient receptive field without occupying huge computational costs, we propose the **M**utual re**C**eption f**LA**re **RE**moval **N**etwork (McLaren) with a novel convolution mechanism named **M**utual **R**eception **Conv**olution (MRConv). The MRConv aggregates multiple convolutional

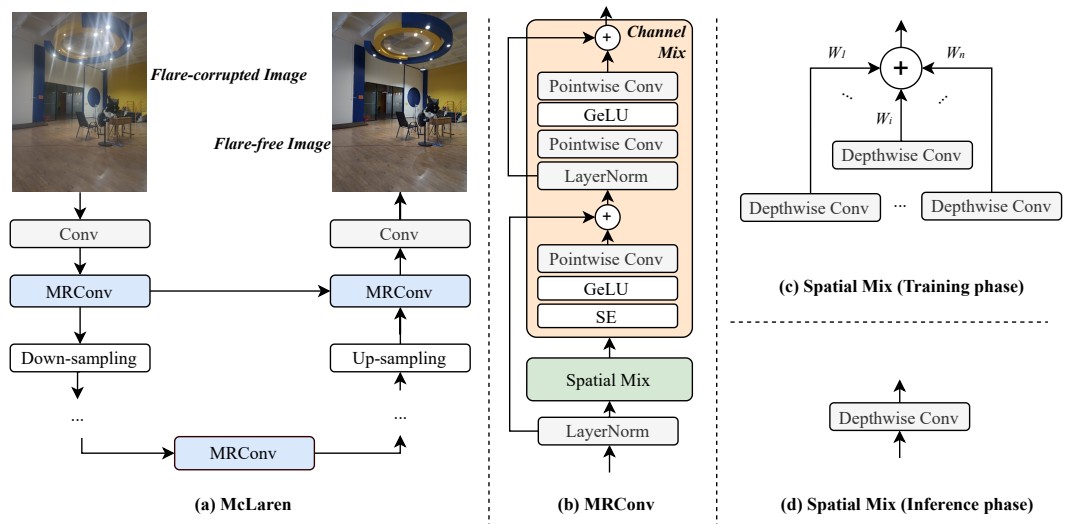

Figure 3: Architecture of our McLaren network. Our McLaren adopts Unet as the backbone and each layer embeds with an MRConv. The MRConv adopts spatial and channel mix to gain reception fields with different sizes. Furthermore, the spatial mix can be further accelerated by the re-parameterization mechanism.

layers with different kernel sizes (from small to large), and fuse them from the perspective of spatial and channel, as compared to transformer-based methods Vaswani et al. (2017), the large kernel convolution has been proved to be efficient with less computational cost Han et al. (2021) and the small kernel convolution helps capture the small-scale patterns Ding et al. (2022). Such a design compels the network to thoroughly analyze flares with arbitrary sizes. We further simplify and accelerate the convolution process with the re-parameterization mechanism Ding et al. (2021).

We conduct extensive experiments that demonstrate that compared with Flare7k++ Dai et al. (2023) dataset, our dataset can help networks gain more robustness to real-world scenarios and our McLaren surpasses state-of-the-art methods with fewer parameters and fewer flops. The contributions of our paper can be summarized as:

- To improve the robustness of existing flare removal networks in real-world scenarios, we propose the FlareReal dataset which contains real-captured image pairs and pure flare images.
- Aiming at thoroughly analyzing and removing the flare, we propose the McLaren network which achieves sufficient receptive field without introducing heavy computational burden.
- We conduct extensive comparisons and ablations experiments to explore functions of our FlareReal dataset and McLaren network.

## 2 RELATED WORK

### 2.1 FLARE REMOVAL DATASET

As a low-level computation vision task, an available training set full of paired data can greatly help the flare removal task. Wu et al. (2021) proposes the first flare removal dataset for daytime flare removal. To handle the nighttime flare, Dai et al. (2022) proposes Flare7k, whose training set contains synthetic nighttime scattering flare images and reflective flare images. As flare images within are all synthetic, to improve the network's ability for real-world scenarios, Dai et al. (2023) extends the Flare7k dataset with real-captured nighttime flare image and names it Flare7k++. Jin et al. (2023) proposes APSF (Atmospheric Point Spread Function)-guided glow rendering method to simulate glow effects from different light sources and a light source aware network to remove glow effects in real-world nighttime hazy images. However, with these datasets in which only flare images are available, all flare-corrupted images in the training data are synthesized, which remain

gaps with real-world scenarios. Zhou et al. (2023) provides a data synthetic resolution to alleviate such phenomenon, which still unavoidable generates artifacts when handling real-captured images. To this end, we propose our FlareReal dataset, which contains real-captured paired images.

## 2.2 FLARE REMOVAL METHODS

When the flare removal dataset is not available, many unsupervised networks have been proposed. Qiao et al. (2021) adopts the idea of cyclegan Zhu et al. (2017) and achieves promising results. They separately detect the light source region and the flare region. The output is generated by blending the flare-removed image and the detected light source mask. However, the generalization of this method is poor and hard to apply in practice. As the proposed flare removal datasets Wu et al. (2021); Dai et al. (2022; 2023), image restoration methods Ronneberger et al. (2015); Zamir et al. (2021; 2022); Wang et al. (2022); Zhang et al. (2023) are capable of generating promising results. However, these methods either lack sufficient receptive field which leads to incomplete flare removal or achieve it with a large computational burden and hardly process high-resolution images. Therefore, we propose our McLaren, which achieves sufficient receptive field in a computational resource-friendly manner. Based on user feedback and observations in real-world environments, we found that nighttime flare is primarily caused by lens smudges, wear and tear, or excessively strong lighting. Therefore, our dataset was primarily collected by intentionally contaminating the lens to simulate these conditions.

## 3 FLAREREAL DATASET

As the synthetic dataset Flare7k++ Dai et al. (2023) provides insufficient robustness for real-world applications, we propose the real-captured paired images dataset FlareReal. Samples of our dataset are shown in Figure 2. We believe that corner cases are essential for dataset construction, especially when the dataset is used for real-world applications. To cover scenarios as many as possible, we collect images from various scenarios. For instance, shapes of light sources in the FlareReal dataset are not fixed (*e.g.*, square, cycle, sphere), which the shape of flares in the Flare7k++ Dai et al. (2023) is always a point light source. Furthermore, the numbers of light sources within captured images are various.

We adopt Samsung Galaxy S23 Ultra, Samsung Galaxy Z Flip5 and Samsung Galaxy Z Fold5 to capture pictures and turn down the scene optimizer to avoid color changes while rendering the raw data into RGB images. To capture the flare-free images and the flare-corrupted images and ensure they contain the same content, we fix the smartphone on a tripod. The flare-corrupted images are captured after we pollute the camera with fingers, dust, oil, or clothes and the flare-free images are captured after we clean the camera with Zeiss lens cleaning tissues. As for the overexposed and underexposed images, we turn our camera into pro mode. In this mode, we set the iso to 800 as a higher iso will introduce unexpected noise. We alter the shutter speed to control the photons within the shooting process.

After obtaining the paired data, we align them for further training. After excluding images with moving objects, we run an image registration method in opencv to align these images and discard unaligned images. Finally, we crop the black area generated during the align process.

However, the data collection process requires large manual efforts and it is nearly impossible to collect millions of image pairs like high-level computer vision task (*e.g.*, Webface260M dataset Zhu et al. (2021) in human face recognition task). Therefore, we also collect flare images from a dark room and synthesize training data with them. Compared with the Flare-R set in the Flare7k++ Dai et al. (2023) dataset, our flare images contain light sources with different colors and different sizes. The resolutions of our flare images are also higher.

## 4 PROPOSED METHOD

After obtaining a real-captured dataset, we focus on finding a network that can fully erase the flare without occupying huge computational resources. Therefore, we propose the **M**utual re**C**eption f**LA**re r**E**moval **N**etwork (McLaren) and depict the structure on Figure 3. Our McLaren network adopts unet Ronneberger et al. (2015) as the backbone and embed each layer of the down-sampling process and the up-sampling process with the proposed **M**utual **R**eception **Conv**olution (MRConv),

which each process contains three layers. This architecture allows for a thorough analysis of flare patterns while maintaining computational efficiency. The flare removal process can be formulated as follows:

$$I^{ff} = f(I^{fc}; \theta) \tag{1}$$

where $I^{ff}$, $I^{fc}$, $f$, and $\theta$ denote the generated flare-free image, the input flare-corrupted image, our Mclaren network, and the training parameter of our network.

### 4.1 MUTUAL RECEPTION CONVOLUTION

The Mutual Reception Convolution (MRConv), as illustrated in fig:Method(b), is a novel convolution mechanism proposed as part of the McLaren network, designed to achieve a sufficient receptive field for analyzing flare patterns without incurring excessive computational costs. MRConv is composed of two key components: the Spatial Mix Block and the Channel Mix Block. These components work in tandem to enable the network to effectively capture and process the diverse scales and patterns of lens flare present in real-world images.

#### 4.1.1 SPATIAL MIX BLOCK

MRConv introduces a Spatial Mix Block to gain different-size reception fields by employing multiple depthwise convolutional layers with various kernel sizes. This approach allows the network to capture flares that spread to arbitrary sizes without incurring excessive computational costs. The spatial mix is further enhanced by trainable weights that allow for the re-parameterization operation, simplifying the convolution process and making it more efficient.

Given image feature $F \in \mathbb{R}^{H \times W \times C}$, the MRConv first normalize it with layer normalization and obtain the normalized version $\overline{F}$. This process can be represented as:

$$\overline{F} = LN(F) \tag{2}$$

where, $H$, $W$, and $C$ denote the width, height, and number of channels of the input features, respectively. $LN(\cdot)$ is the layer normalization. Then, we introduce the spatial mix to gain different-size reception fields. Specifically, the spatial mix employs $n$ depthwise convolutional layers whose kernel sizes are various to obtain features $F_0, \ldots, F_n$ for flares that spread to arbitrary sizes. The smaller kernels capturing fine-grained details and larger kernels capturing broader patterns. Afterward, we sum these features with trainable weights $W_0, \ldots, W_n$. The spatial mix training process can be represented as:

$$F_{spm} = \sum_{i=0}^{n} W_i \cdot DW_i(\overline{F}) \tag{3}$$

where $DW_i(\cdot)$ denotes the depthwise convolutional, $F_{spm}$ represents the output of the spatial mix module. During inference, the Spatial Mix Block leverages a re-parameterization Ding et al. (2021) mechanism to simplify the computation. Small kernel convolutions are effectively transformed into larger kernel convolutions through zero-padding, reducing the number of operations while preserving the receptive field. The spatial mix inference process can be represented as:

$$F_{spm} = DW(\overline{F})) \tag{4}$$

#### 4.1.2 CHANNEL MIX BLOCK

The Channel Mix Block is responsible for capturing the channel-wise dependencies within the feature maps, enhancing the network's ability to understand the relationships between different features and improve the overall representation for flare removal. We introduce the Squeeze-and-Excitation mechanism (SE) Hu et al. (2018) to compute the weight of each channel and enhance the useful information by explicitly exploring the dependency between these features. Following the SE block, a GeLU Hendrycks & Gimpel (2016) activation function is applied, followed by a pointwise convolutional layer to fuse the channel features. This fusion allows the network to integrate the channel-wise information, resulting in a more comprehensive feature representation. To avoid gradient disappearance and to promote the flow of information, the output of the Channel Mix Block

| Training set | Method | Params (M) | Macs (G) | Flare7k++ | | | | | FlareReal | | |
|---|---|---|---|---|---|---|---|---|---|---|---|
| | | | | PSNR | G-PSNR | S-PSNR | SSIM | LPIPS | PSNR | SSIM | LPIPS |
| Flare7k++ | Unet | **9.0** | 62.36 | 27.189 | 23.527 | 22.907 | 0.894 | 0.0452 | 18.545 | 0.555 | 0.2024 |
| | HINet | 88.6 | 683.32 | 27.548 | 24.081 | 22.907 | 0.892 | 0.0464 | 19.125 | **0.573** | 0.2134 |
| | Uformer | 20.4 | 162.09 | 27.633 | 23.949 | 22.603 | 0.894 | 0.0428 | 18.249 | 0.558 | 0.1991 |
| | McLaren (Ours) | 13.2 | **33.10** | **27.736** | **24.193** | **23.256** | **0.900** | **0.0416** | **19.348** | 0.570 | **0.1937** |
| FlareReal | Unet | **9.0** | 62.36 | 26.825 | 24.005 | 22.895 | 0.885 | 0.0410 | 19.712 | 0.584 | 0.1952 |
| | HINet | 88.6 | 683.32 | 28.217 | 25.237 | 24.712 | **0.910** | **0.0350** | 20.725 | **0.600** | 0.1856 |
| | Uformer | 20.4 | 162.09 | 27.276 | 24.285 | 23.464 | 0.905 | 0.0385 | 20.632 | 0.598 | **0.1837** |
| | McLaren (Ours) | 13.2 | **33.10** | **28.245** | **25.780** | **24.756** | 0.903 | 0.0355 | **20.748** | 0.595 | 0.1861 |
| Flare7k++ & FlareReal | Unet | **9.0** | 62.36 | 27.726 | 24.753 | 23.961 | 0.887 | 0.0394 | 20.499 | 0.595 | 0.1882 |
| | HINet | 88.6 | 683.32 | 28.737 | 25.680 | 25.074 | 0.911 | **0.0341** | 20.389 | 0.595 | 0.1867 |
| | Uformer | 20.4 | 162.09 | 28.366 | 25.236 | 24.261 | 0.908 | 0.0359 | 20.499 | 0.596 | 0.1875 |
| | McLaren (Ours) | 13.2 | **33.10** | **28.778** | **25.795** | **25.234** | **0.914** | 0.0347 | **20.989** | **0.603** | **0.1826** |

Table 1: Quantitative results about our dataset and our method. The best results have been shown in **bold**.

is added back to the input feature maps, similar to a residual connection. The formula for this process is given by:

$$\hat{F} = PC(GeLU(SE(F_{spm}))) + F \tag{5}$$

where $SE(\cdot)$, $GeLU(\cdot)$ and $PC(\cdot)$ denote the SE operator, GeLU activation function and a pointwise convolutional layer, respectively. Then, we introduce a Convolutional MLP (ConvMLP) block to further enhance the mutual information among features with various receptive fields. The ConvMLP block initiates with a pointwise convolutional layer, which is a convolution operation with a kernel size of $1 \times 1$. Following the pointwise convolution, a GeLU activation function is applied to learn more complex patterns and relationships within the data. The output of the GeLU activation is then passed through another pointwise convolutional layer to adjust and fine-tune the integrated features based on the learned representations from the previous layers. The mathematical representation of the ConvMLP block can be expressed as:

$$\hat{F}_{LN} = LN(\hat{F}) \tag{6}$$

$$F_{chm} = PC_2(GeLU(PC_1(\hat{F}_{LN}))) + \hat{F} \tag{7}$$

where $F_{chm}$ is the output feature after the ConvMLP block (or Channel Mix Block), $PC_1(\cdot)$ and $PC_2(\cdot)$ represent the pointwise convolutional layers.

Our McLaren network combines the L1 loss and the perceptual loss Johnson et al. (2016) between the generated flare-free image and the ground truth flare-free image as the loss function. This combination ensures that the network not only minimizes the pixel-wise error but also preserves the perceptual quality of the restored images. The loss function can be represented as:

$$L = \lambda_1 \cdot L1(I_{ff}, I_{gt}) + \lambda_2 \cdot LP(I_{ff}, I_{gt}) \tag{8}$$

where $L1(\cdot)$ is the L1 loss, $LP(\cdot)$ is the perceptual loss, $I_{ff}$ is the generated flare-free image, $I_{gt}$ is the ground truth flare-free image, and $\lambda_1$ and $\lambda_2$ are the weights for the respective losses.

## 5 EXPERIMENTS

### 5.1 IMPLEMENTATION DETAILS

For the training set, we employ the Flare7k++ dataset Dai et al. (2023) and our FlareReal dataset and both datasets adopt the same synthesize strategy with images from Flickr24k dataset Zhang et al. (2018) as our dataset also contains flare images. We adopt the Adam optimizer to optimize our McLaren network with 200 epochs. The learning rate is set to 1e-4.

As for the testing set, the Flare7k++ dataset and our FlareReal dataset are employed. Moreover, to test the robustness of our dataset and our method, we also introduce images from Zhou et al. (2023) to testify whether our dataset and method can handle images with unseen flare. Following the experimental setting of Flare7k++ dataset Dai et al. (2023), we employ PSNR, G-PSNR, S-PSNR, SSIM, and LPIPS for Flare7k++ dataset and PSNR, SSIM, and LPIPS for our FlearReal dataset, as image within our dataset are captured from real and contains no annotations about glare and streak.

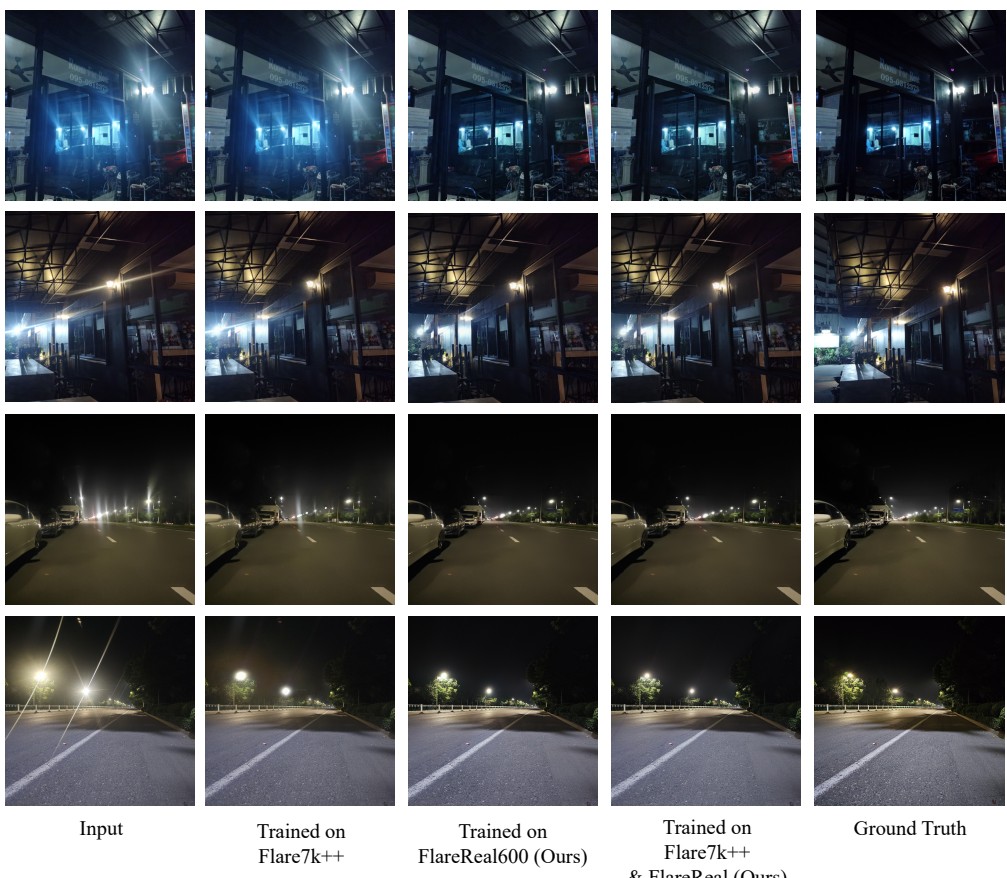

Input          Trained on          Trained on          Trained on          Ground Truth
               Flare7k++          FlareReal600 (Ours)    Flare7k++
                                                       & FlareReal (Ours)

Figure 4: Qualitative results about datasets. The testing method is our McLaren. Images on the first two rows are from Flare7k++ real set and the remaining images are from our FlareReal validation set.

## 5.2 COMPARISON EXPERIMENTS

To validate the functions of our dataset and network, we follow the setting of Flare7k++ Dai et al. (2023) and employ original Unet Ronneberger et al. (2015), HINet Chen et al. (2021), and Uformer Wang et al. (2022) as baselines. We discard the Restormer method Zamir et al. (2022), as the resolutions of images on our validation set are 4k and restormer hardly handles $512 \times 512$ images on Nvidia A100 GPU. Each MRConv layer in our McLaren employs four depthwise convolutional layers whose kernel sizes are 9, 7, 5, and 3.

### 5.2.1 FLAREREAL VS FLARE7K++

Aiming at finding out how to gain the highest score for current flare removal methods, we train these networks three times: networks trained on Flare7k++ dataset, networks trained on our FlareReal dataset, and networks trained on the mixed dataset which combined both datasets. We show the quantitative and qualitative results in tab:com and fig:mclaren, and fig:robust.

As shown in tab:com, for Flare7k++ real set, methods trained on our dataset approach surpass themselves trained on Flare7k++ dataset (-0.364dB PSNR gains on unet, 0.669dB PSNR gains on HINet, -0.357dB PSNR gains on uformer, 0.509 PSNR gains on our McLaren). For our real-captured FlareReal dataset, all methods trained on our dataset significantly surpass those trained on Flare7k++ dataset (1.167dB PSNR gains on unet, 1.600dB PSNR gains on HINet, 2.383 PSNR gains on former, 1.400 PSNR gains on our McLaren). Moreover, the method trained on the mixed dataset which combines both the Flare7k++ dataset and our dataset achieves the highest scores on Flare7k++ dataset. Such results demonstrate that our dataset helps current image restoration methods to deal with real-world scenarios.

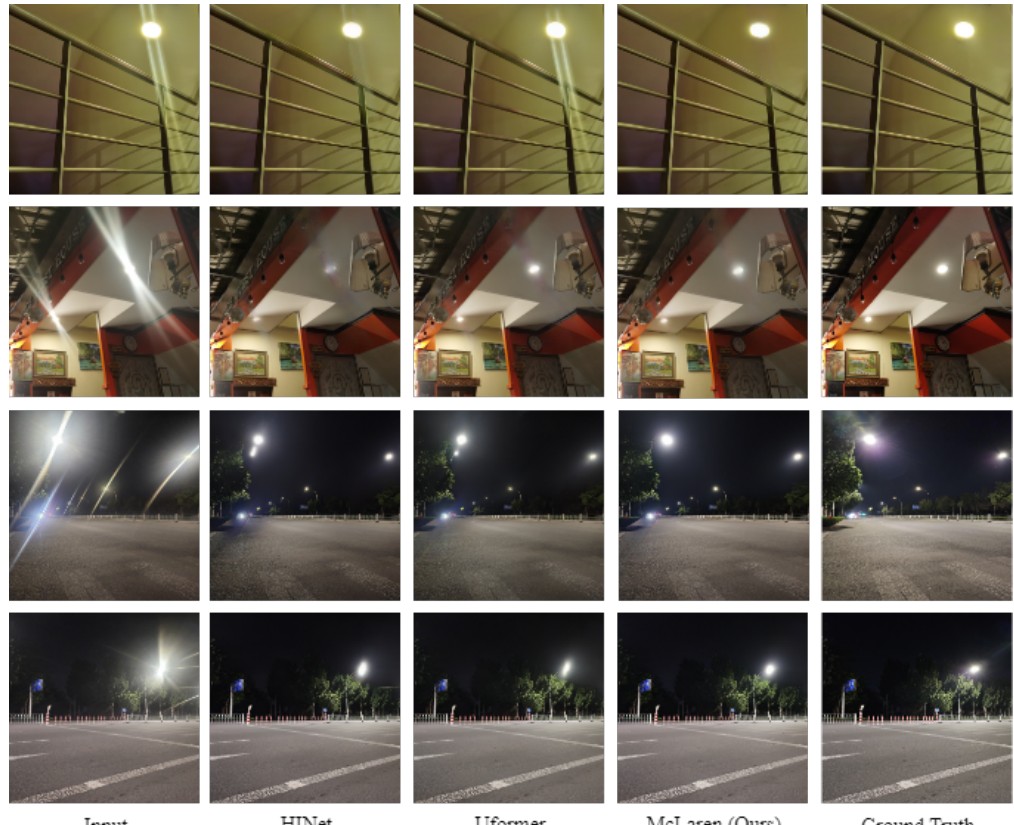

Figure 5: Qualitative results about methods. Images in the first two rows are from Flare7k++ dataset Dai et al. (2023) and remain images are from our FlareReal dataset.

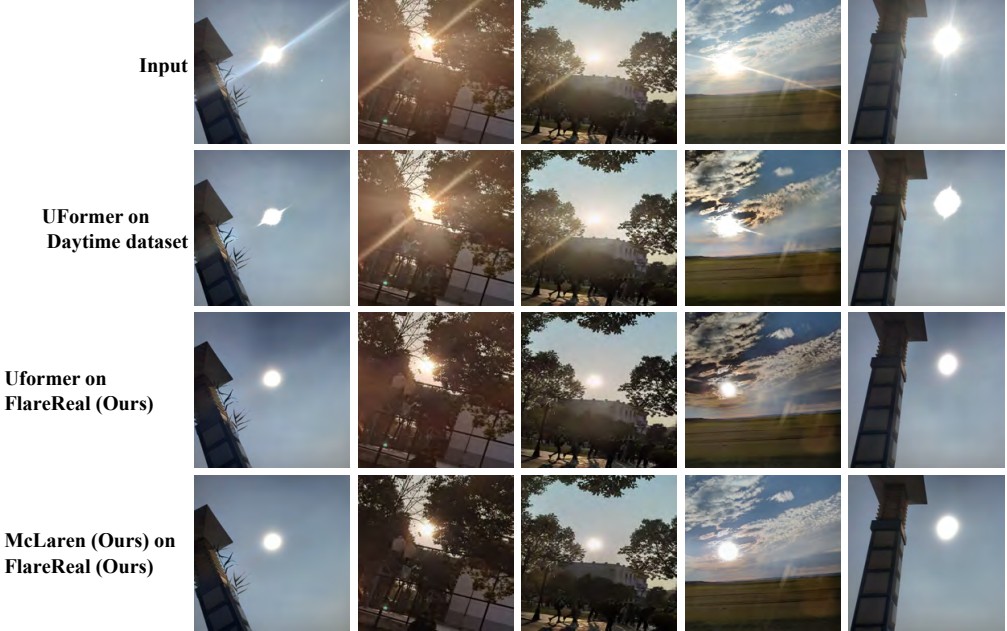

Figure 6: Qualitative results on unseen flare (natural sunlight) Zhou et al. (2023). Zoom in for a better view.

From the perspective of visual quality, according to fig:mclaren, Uformer and our McLaren trained on our FlareReal dataset both remove more flare and generate fewer artifacts than themselves trained on Flare7k++ dataset. Moreover, from fig:robust, our McLaren and Uformer trained on our dataset

| Method | PSNR | G-PSNR | S-PSNR | SSIM | LPIPS |
|---|---|---|---|---|---|
| 753 | 27.537 | 23.982 | 22.664 | 0.893 | 0.0442 |
| 9753 *w/o* W | 27.708 | 24.112 | 23.070 | 0.898 | 0.0418 |
| 9753 | **27.736** | **24.193** | **23.256** | **0.900** | **0.0416** |

Table 2: Ablation results about our McLaren. 753 represents the kernel sizes of the depthwise convolution in the spatial mix are 7, 5, and 3. Correspondingly, 9753 represents kernel sizes are 9, 7, 5, and 3. *w/o* W means without trainable weight in the spatial mix. The best results have been shown in **bold**.

which contains nighttime flare handle the flare much better than Uformer trained on the current daytime flare dataset Wu et al. (2021), which demonstrates that networks trained on our dataset are capable of handling unseen flare scenarios.

### 5.2.2 MCLAREN

To explore the functions of our McLaren. we show the quantitative and qualitative results on tab:com and fig:robust and fig:com. From tab:com, our method adopts nearly the fewest parameters and Macs to achieve the highest PSNR, G-PSNR, and S-PSNR scores and mostly achieves the best SSIM and LPIPS scores.

As for the visual quality, our method generates the least artifacts while preserving the light source without distortion. Given the image from the first row of fig:com, our method manages to fully remove the flare without generating any artifacts. As for the image on the second row, our method introduces the least artifacts and preserves the light source while removing the strong streak around the light source. For images from the last two rows, our method manages to recover the shape of the light source, while HINet and Uformer fail. Such results prove that our McLaren manages to remove the flare with limited computational cost. The McLaren network's ability to handle unseen flare scenarios, as demonstrated by the qualitative results on natural sunlight flare images, shows that the network is not only effective on the training data but also generalizes well to new flare conditions. This is a testament to the network's robustness and adaptability, which is superior to the Uformer and Unet, as shown in fig:robust.

### 5.3 ABLATION EXPERIMENTS

#### 5.3.1 COMPARISON RESULTS OF KERNEL SIZE IN MRCONV

The McLaren network introduces the Mutual Reception Convolution (MRConv), which aggregates multiple convolutional layers with different kernel sizes to achieve a sufficient receptive field without incurring excessive computational costs. This design allows the network to analyze flare patterns of arbitrary sizes effectively, which is a significant advantage over the Uformer and Unet, especially for high-resolution images. We try to analyze the design of our spatial mix of the MRConv. We explore the functions of hyper-parameters of the kernel sizes and trainable weights we introduce to fuse the depthwise convolutions. We show the results in tab:ablation. The training and testing datasets are both Flare7k++ dataset. According to tab:ablation, the method with larger receptive fields gains 0.199dB, 0.211dB, and 0.592dB promotions on PSNR, G-PSNR, and S-PSNR scores. Moreover, the trainable weights bring 0.028dB, 0.081dB, and 0.186dB promotions on PSNR, G-PSNR, and S-PSNR scores.

## 6 CONCLUSION

In this paper, we propose a real-capture dataset named FlareReal and a novel flare removal network named McLaren. The FlareReal alleviates the issue that current methods trained on Flare7k++ dataset lacks robustness to real-world applications by providing 3,000 real-captured image pairs and 500 flare images with more colors. Moreover, as current methods either lack sufficient receptive field to fully analyze the flare pattern or require huge computational costs that hardly handle high-resolution images, the McLaren network proposes the MRConv which adopts multiple depthwise convolutions with various kernel sizes and aggregates them from the perspective of spatial and channel. Extensive results prove the superiorities of FlareReal dataset and McLaren network.

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

# A  ADDITIONAL ABLATION STUDY

## A.0.1  COMPARISON RESULTS WITH PSF-BASED METHOD

We also conducted a comparative analysis with the Point Spread Function (PSF)-based flare generation method Jin et al. (2023) to evaluate the efficacy of our real-captured dataset, as shown in fig:1002. This evaluation was performed by scrutinizing the discrepancies between our real-collected images and those artificially synthesized using the PSF technique on pristine images extracted from our validation dataset. Our analysis revealed that the real-collected images exhibited a more diverse array of flare patterns than PSF method, including shimmer, halo, and streak effects. These patterns are indicative of the complex optical interactions that occur in real-world scenarios, which are not fully replicated by the PSF-based synthesis approach. Furthermore, upon meticulous observation of the synthesized images, it was identified that the PSF method erroneously detected a lane within the imagery as a light source, consequently imparting flare artifacts onto it. This finding underscores the limitations of the PSF-based method in accurately simulating the intricacies of real-world flare phenomena, particularly its propensity to misidentify non-light sources as flare origins. This misclassification not only affects the authenticity of the synthesized flare-corrupted images but also poses a significant challenge for flare removal algorithms that rely on such synthetic data for training and validation purposes.

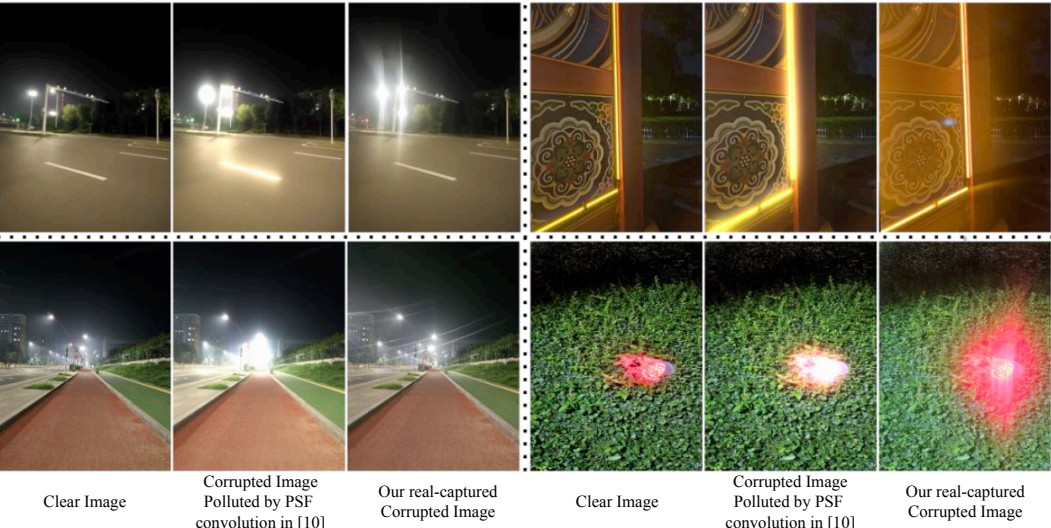

| Clear Image | Corrupted Image Polluted by PSF convolution in [10] | Our real-captured Corrupted Image | Clear Image | Corrupted Image Polluted by PSF convolution in [10] | Our real-captured Corrupted Image |

Figure 7: Comparison between images generated by PSF method in Jin et al. (2023) and our real-captured images. Our images contains more types of flares without changing the color of the light source

## A.0.2  GENERALIZATION TO REAL-WORLD FLARE-LIKE ARTIFACTS

As illustrated in Figure 8, our McLaren model demonstrates superior flare suppression performance in both scenarios, despite never having been explicitly trained on such artifacts. On the RealNightHaze dataset Jin et al. (2023), which contains strong scattering glare introduced by atmospheric particles, our model effectively attenuates the flare while preserving the structural integrity and intensity of true light sources. This indicates the model's capacity to generalize beyond lens-induced artifacts to more complex, medium-induced scattering phenomena. Similarly, on the Dataset in Sharma & Tan (2021), which includes high-dynamic-range light blooms and reflections, the McLaren network is capable of reducing light-induced distortions while avoiding over-suppression of non-flare regions. These results provide compelling evidence that models trained on FlareReal possess enhanced robustness in real-world scenarios and exhibit strong cross-task generalization.

# B  ADDITIONAL VISUAL RESULTS

In this part, we provide additional visual results compared with  Dai et al. (2023).

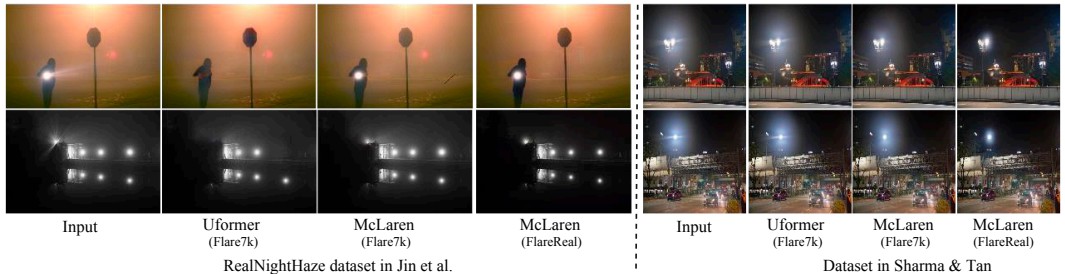

| Input | Uformer
(Flare7k) | McLaren
(Flare7k) | McLaren
(FlareReal) | Input | Uformer
(Flare7k) | McLaren
(Flare7k) | McLaren
(FlareReal) |
| RealNightHaze dataset in Jin et al. | | | | Dataset in Sharma & Tan | | | |

Figure 8: Visual quality comparison on datasets in Jin et al. (2023) and Sharma & Tan (2021). Our McLaren network manages to eliminate all the flare without removing the light source or introducing any artifacts. All methods are trained on mixed dataset (flare7k++ dataset & FlareReal dataset (Ours)).

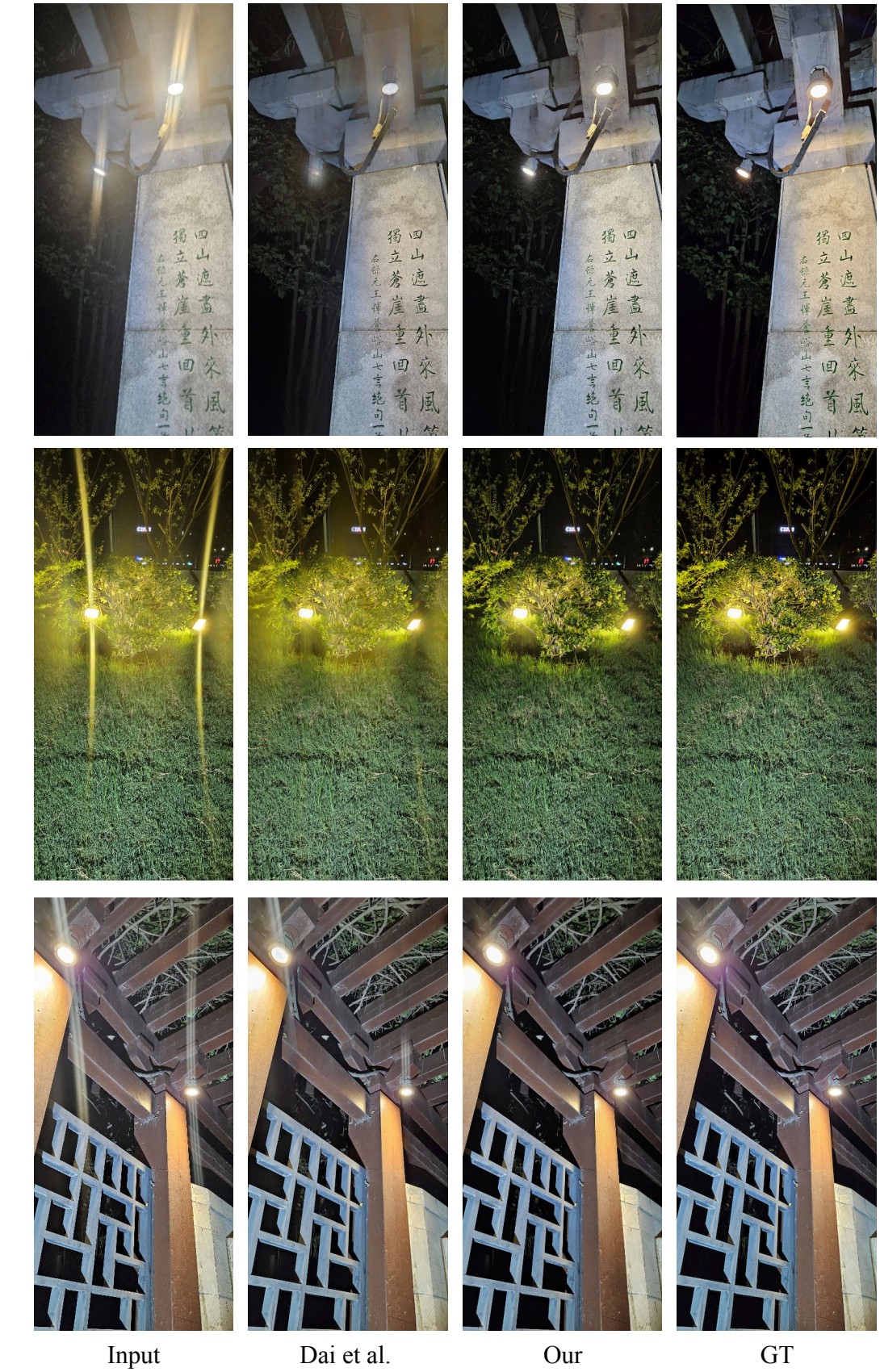

Figure 9: Visual results achieved by different methods on the FlareReal validation dataset.

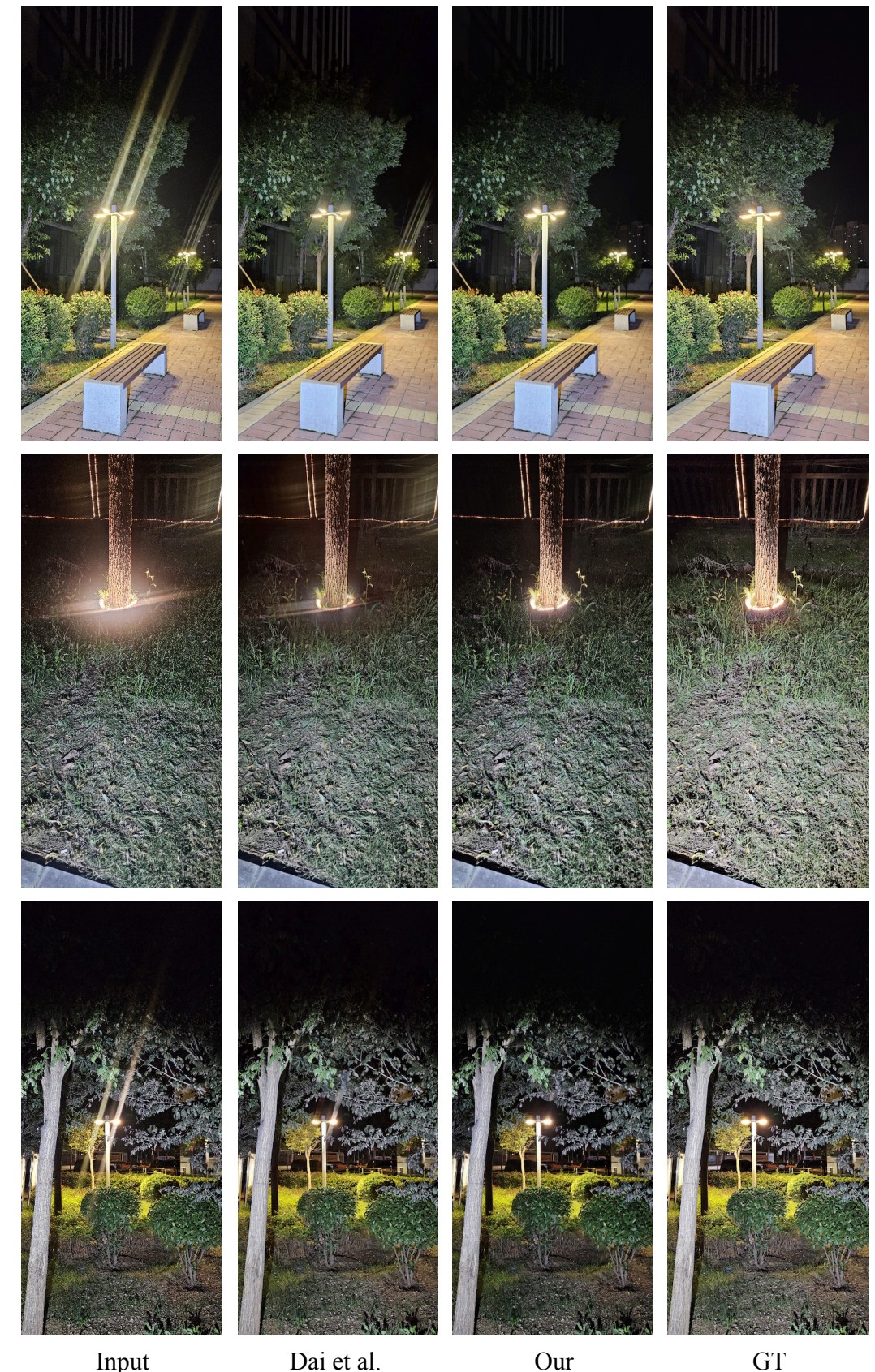

Input    Dai et al.    Our    GT

Figure 10: Visual results achieved by different methods on the FlareReal validation dataset.

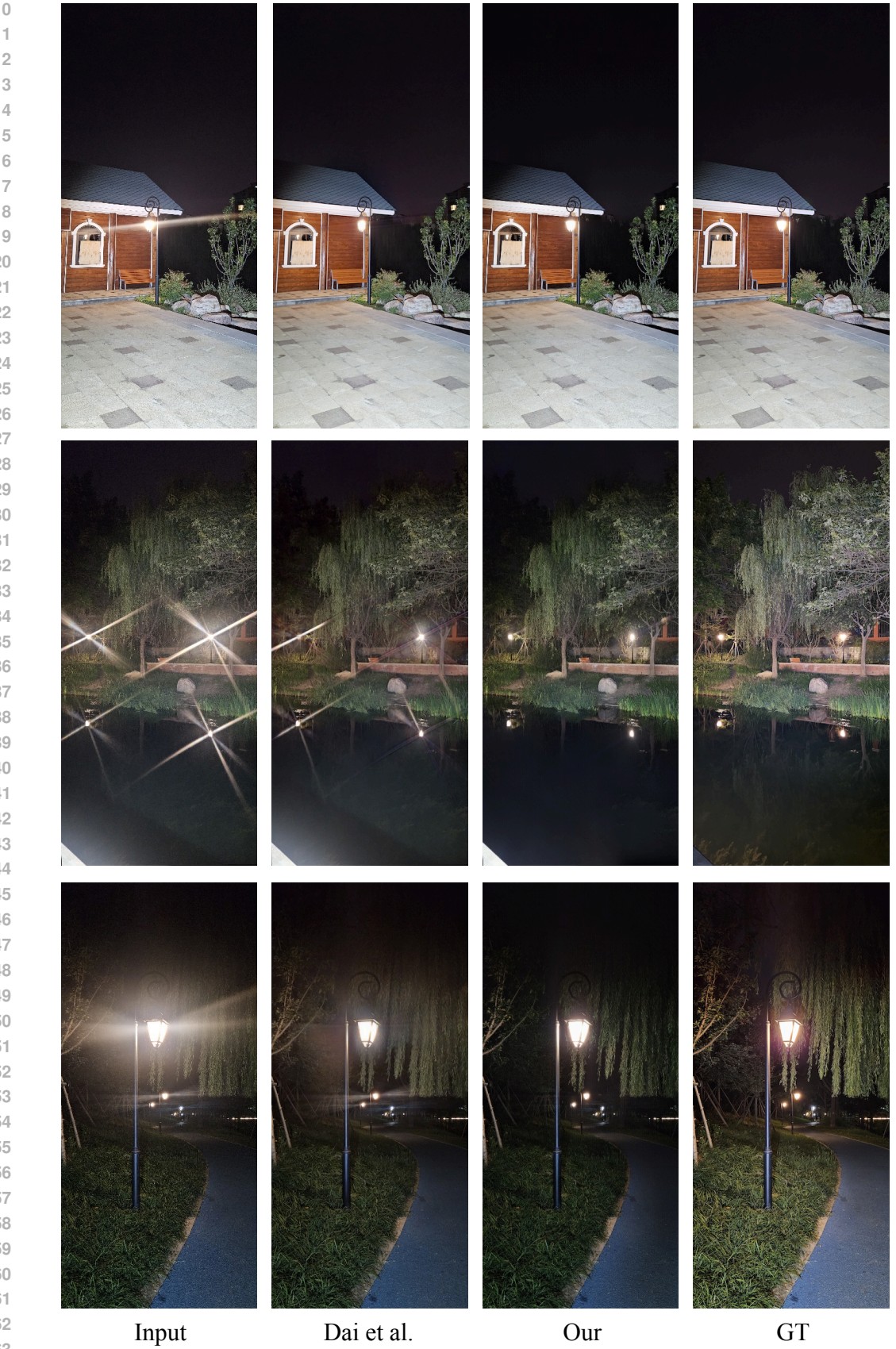

Figure 11: Visual results achieved by different methods on the FlareReal validation dataset.

