

| Input | Flare7K | Flare7K++ | FlareX | Ours | GT |

Figure 1: Visual comparison of flare removal on the Flare7k real test set. The name of each column represents the dataset that is used to train Uformer.

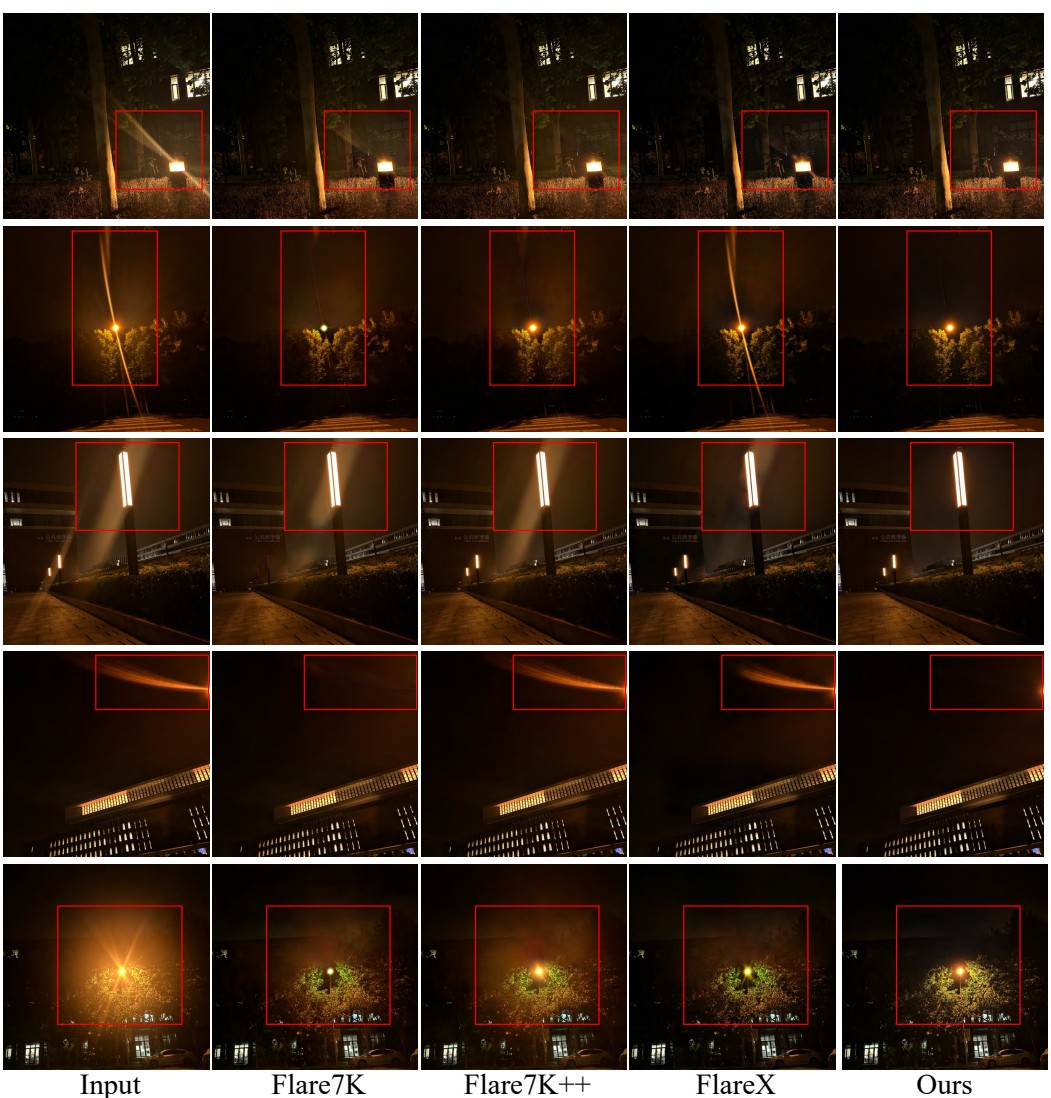

| Input | Flare7K | Flare7K++ | FlareX | Ours |

Figure 2: Visual comparison of flare removal on the FlareX test set. The name of each column represents the dataset that is used to train Uformer.

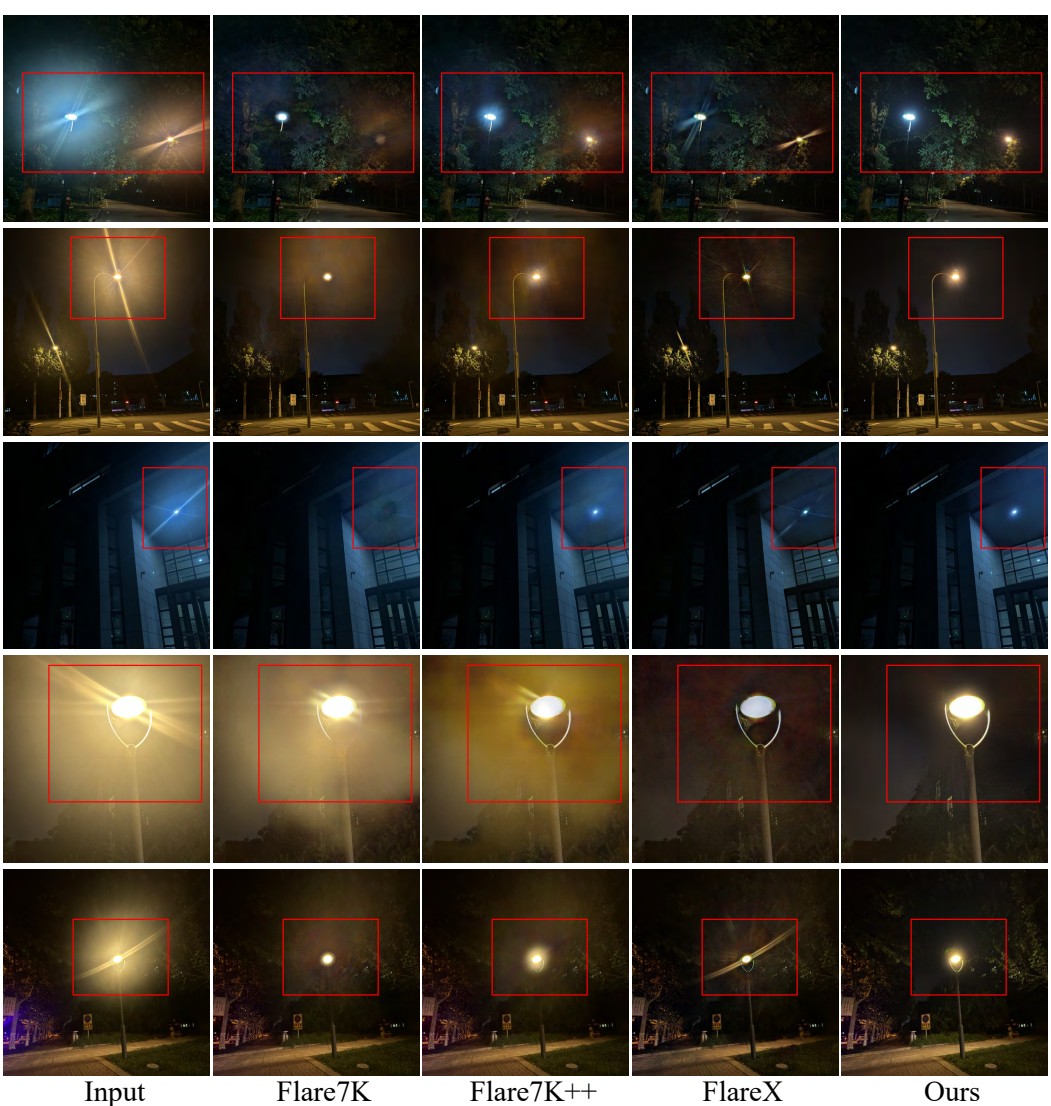

| Input | Flare7K | Flare7K++ | FlareX | Ours |
|-------|---------|-----------|--------|------|

Figure 3: Visual comparison of flare removal on the FlareX test set. The name of each column represents the dataset that is used to train Uformer.

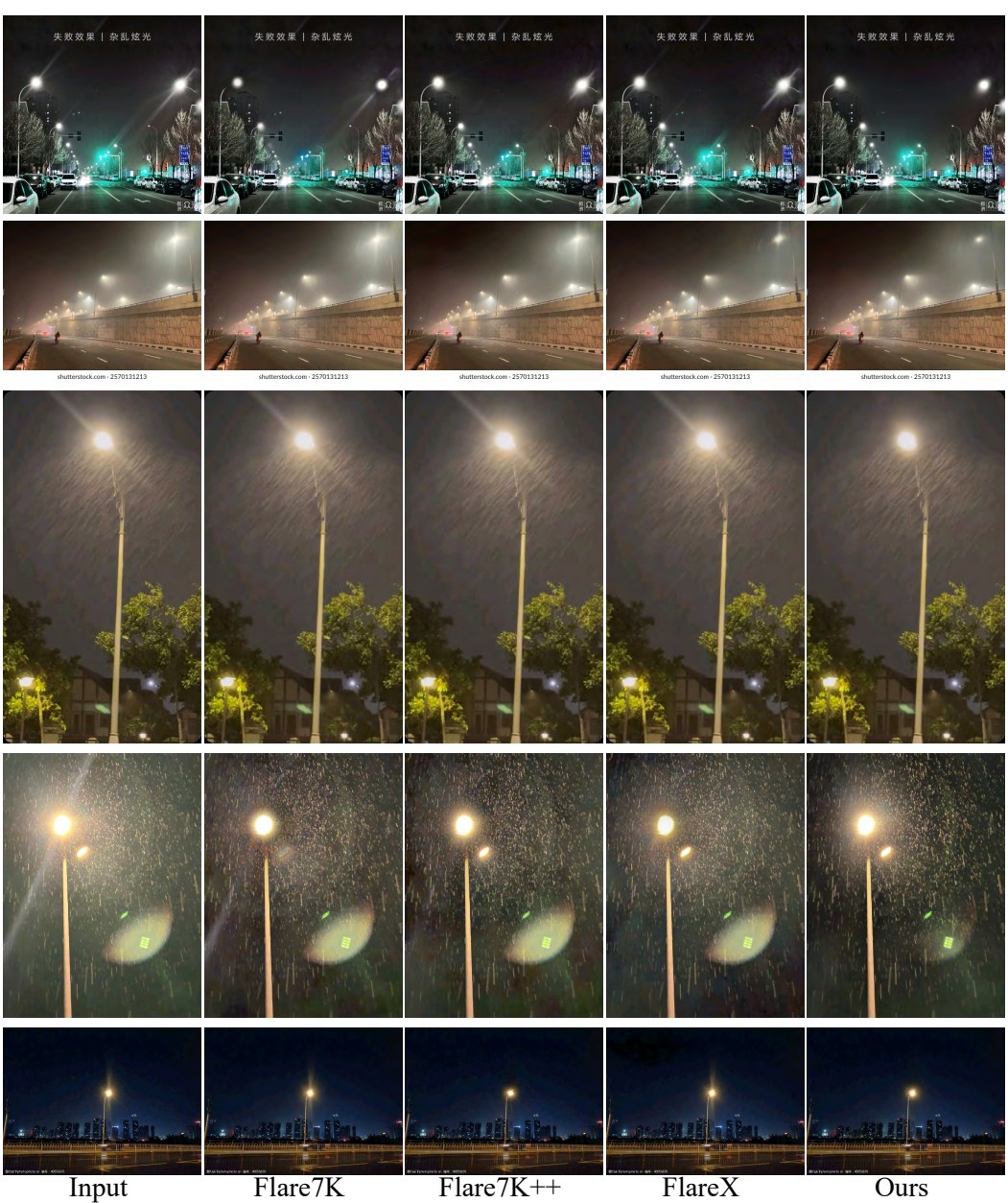

Figure 4: Visual comparison of flare removal on the test set collected from Google Image Search. The name of each column represents the dataset that is used to train Uformer.

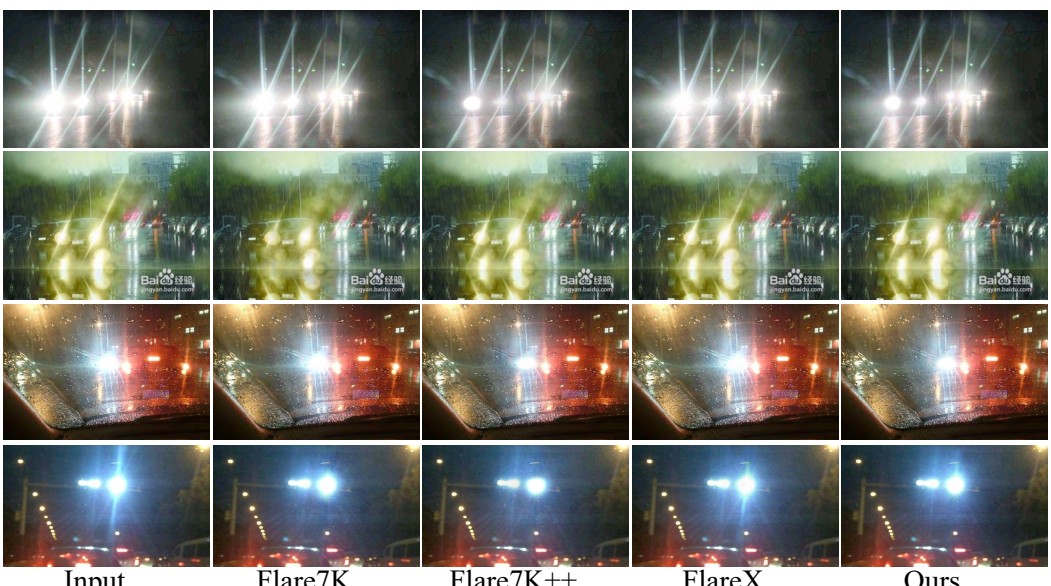

| Input | Flare7K | Flare7K++ | FlareX | Ours |

Figure 5: Visual comparison of flare removal on the test set collected from Google Image Search. The name of each column represents the dataset that is used to train Uformer.

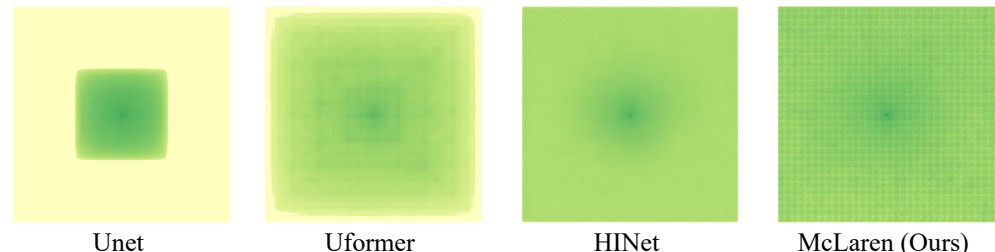

| Unet | Uformer | HINet | McLaren (Ours) |

Figure 6: The Effective Receptive Field (ERF) visualization for Unet, Uformer, HINet and our McLaren.