# OpenReview forum: "Real-Captured Paired Dataset for Nighttime Flare Removal"
_ICLR.cc/2026/Conference — ICLR 2026 Conference Withdrawn Submission_

### Official Review · Reviewer_541v · 2025-10-31

**Soundness:** 4
**Presentation:** 3
**Contribution:** 3
**Rating:** 6
**Confidence:** 3

**Summary:**

The paper proposes a new dataset, FlareReal, which is designed to enhance the robustness of flare removal models for real-world scenarios. The authors introduce a McLaren network to process these images efficiently, combining large and small convolution kernels to achieve sufficient receptive fields without high computational costs. Extensive experiments demonstrate that the FlareReal dataset improves existing methods' performance and that McLaren outperforms other state-of-the-art models with fewer parameters.

**Strengths:**

The gap between synthetic data and real-world images is a critical and persistent challenge in low-level vision. This paper provides a direct and practical solution for the task of flare removal, which is an important problem in real-world scenarios.

**Weaknesses:**

(1)The FlareReal validation set is small, consisting of only 50 image pairs. A larger validation set would provide greater statistical confidence.
(2)The baseline methods tested are limited to UNet, HINet, and Uformer. The authors can explore additional baseline methods (such as MPRNet and Restormer) and select one or two recent state-of-the-art methods for dataset validation.
(3)To advance progress in this field, it is recommended to include a dataset download link for community access once the paper is accepted.

**Questions:**

(1)Will this dataset be open-sourced?
(2)Would McLaren’s computational efficiency hold up in more resource-constrained environments, like edge devices or mobile applications? This is an issue that needs to be considered in practical applications.
(3)Are there any notable failure cases or types of flares where the method still struggles? For example, extremely severe flares that occlude most of the image, or complex, colorful artifacts?
(4)The design of MRConv is similar to the paper “Reparameterized Multi-Resolution Convolutions for Long Sequence Modelling”. Please explain the differences between the two designs.

Paper: Cunningham, Jake, et al. "Reparameterized Multi-Resolution Convolutions for Long Sequence Modelling." Advances in Neural Information Processing Systems 37 (2024): 27121-27152.

---

> ### Author Response · Authors · 2025-11-24
>
> We sincerely thank you for their valuable comments and for highlighting specific areas where our validation and analysis can be strengthened. We appreciate the opportunity to clarify the robustness of our evaluation and the specific architectural distinctions of our method. Below, we provide point-by-point responses to the concerns raised.
>
> W1. The FlareReal validation set is small, consisting of only 50 image pairs. A larger validation set would provide greater statistical confidence.
>
> > We agree with you that a larger validation set improves statistical confidence. We have made the source code and a subset of the dataset publicly available, including 600 training pairs and 50 validation pairs. Furthermore, we have also held CVPR workshops and competitions. In the initial stage of the competition, we released a verification set of 100 pairs of images, but the participants agreed that the storage space occupied was too large, so we reduced the number of verification sets from 100 pairs to 50 pairs. In the final revision, we wii increase the validation set size to 100 pairs to ensure more rigorous evaluation.
>
> W2. The baseline methods tested are limited to UNet, HINet, and Uformer. The authors can explore additional baseline methods (such as MPRNet and Restormer) and select one or two recent state-of-the-art methods for dataset validation.
>
> > We agree that including more recent baselines such as MPRNet and Restormer can strengthen the comparison. We will add the them in the final revision.
> >
> > Following your suggestion and further evaluate cross-sensor generalization, we conducted additional experiments on the **FlareX** test set, which **contains images captured using both professional cameras and smartphones**. Our model trained on FlareReal consistently outperforms models trained on Flare7K++ or FlareX [1], demonstrating stronger robustness across heterogeneous optical systems. For visual comparison results in Flare7K and FlareX test set, please refer to Figure 1, 2 and 3 in rebuttal.pdf of the Supplementary Material.
> >
> > |       Dataset       |  Uformer   |  Uformer  |  Uformer  |
> > | :-----------------: | :--------: | :-------: | :-------: |
> > | **FlareX test set** |  **PSNR**  | **SSIM**  | **LPIPS** |
> > |      Wu et al       |     -      |   0.608   |   0.143   |
> > |       Flare7K       |   37.771   |   0.633   |   0.139   |
> > |      Flare7K++      |   37.448   |   0.631   |   0.142   |
> > |       FlareX        |   39.292   |   0.658   |   0.133   |
> > |         Our         | **40.219** | **0.659** | **0.126** |
> >
> > In addition, we also tested our model on user-uploaded flare images collected from Google Image Search, including images captured under rainy, foggy, and low-visibility conditions. As these images come from unknown devices, the camera type (smartphone, DSLR, action camera, dashcam, etc.) cannot be determined (for visual comparison results, please refer to Figure 4 in rebuttal.pdf of the Supplementary Material). Nevertheless, our model trained on FlareReal still performed better than models trained on Flare7K++ and FlareX. This empirical evidence suggests that although FlareReal is collected primarily using smartphone cameras, it captures flare characteristics that transfer well to different imaging devices and optics, thereby reflecting strong dataset robustness. In the revision, we will include these quantitative results on FlareX and add a dedicated paragraph discussing cross-device generalization.
>
> W3 & Q1. To advance progress in this field, it is recommended to include a dataset download link for community access once the paper is accepted. Will this dataset be open-sourced?
>
> > We have made the source code and a subset of the dataset publicly available, including 600 training pairs and 50 validation pairs. Furthermore, we have also held CVPR workshops and competitions. Our full FlareReal dataset will be publicly released upon acceptance.
>
> Q2. Would McLaren’s computational efficiency hold up in more resource-constrained environments, like edge devices or mobile applications? This is an issue that needs to be considered in practical applications.
>
> > Yes, McLaren is specifically designed for such environments. Our MRConv uses depthwise kernels for all multi-scale receptive fields. Depthwise operations are extremely efficient on mobile SoCs. As shown in Table 1, McLaren achieves SOTA performance with only 33.10 G MACs, which is drastically lower than HINet (683.32 G) and Uformer (162.09 G). This low FLOPs count makes it highly suitable for mobile deployment. Furthermore, our method has been commercially deployed on smartphones featuring Qualcomm Snapdragon 8 Elite, delivering an inference time of 687 ms on the CPU for an input resolution of 1024×1024.

---

> ### Author Response · Authors · 2025-11-24
>
> Q3. Are there any notable failure cases or types of flares where the method still struggles? For example, extremely severe flares that occlude most of the image, or complex, colorful artifacts?
>
> > To further assess real-scene generalization, we also tested our model on user-uploaded flare images collected from Google Image Search, including images captured under rainy, foggy, and low-visibility conditions. As these images come from unknown devices, the camera type (smartphone, DSLR, action camera, dashcam, etc.) cannot be determined (for visual comparison results, please refer to Figure 4 in rebuttal.pdf of the Supplementary Material). Nevertheless, our model trained on FlareReal still performed better than models trained on Flare7K++ and FlareX, demonstrating that the flare priors learned from FlareReal generalize well to unseen weather conditions and unknown camera types. We acknowledge that there remain failure cases, particularly under extremely harsh conditions where flare is overwhelmingly strong or when the scene contrast is severely degraded (e.g., heavy rain + fog + multiple glare sources). These bad cases occur not only for our model, but also for models trained on Flare7K++ and FlareX. For visual comparison results, please refer to Figure 5 in rebuttal.pdf of the Supplementary Material. We will add more such instances in the revision.
>
> Q4. The design of MRConv is similar to the paper “Reparameterized Multi-Resolution Convolutions for Long Sequence Modelling”. Please explain the differences between the two designs.
>
> > Thank you for pointing out this related work. While the names are similar, the domains and architectural goals are fundamentally different: A. The referenced work is designed for 1D long-sequence modeling. Our MRConv is designed for 2D spatial modeling specific to flare patterns, integrating both spatial and channel dependencies. B. The referenced method uses multi-resolution kernels primarily for temporal expansion. MRConv uses multi-scale depthwise spatial kernels with: trainable fusion weights and 2D re-parameterization into a single large depthwise kernel. C. MRConv contains: Spatial Mix, SE-based channel weighting and ConvMLP refinement. This creates bidirectional mutual reception between spatial kernels and channel dependencies, absent in this related work. D. We organized the CVPR competition in early 2024, and our ideas have been initially formed, which is work at the same time as "Reparameterized Multi-Resolution Convolutions for Long Sequence Modelling."

---

### Official Review · Reviewer_mEUy · 2025-10-31

**Soundness:** 2
**Presentation:** 1
**Contribution:** 1
**Rating:** 2
**Confidence:** 4

**Summary:**

This paper tackles the task of flare removal in images, noting that most existing methods rely on synthetic datasets like Flare7k, which limits real-world generalization. To address this, the authors introduce FlareReal, a new dataset containing 3,000 real-captured paired images and 500 standalone flare samples. They also propose McLaren, a UNet-based network incorporating a new Mutual Reception Convolution (MRConv) module that combines multi-kernel depthwise convolutions with re-parameterization for efficient receptive field expansion. Experiments on Flare7k++ and FlareReal show modest improvements over existing baselines such as UNet, HINet, and Uformer.

**Strengths:**

1. Introduction of a new real-captured dataset:
The paper introduces FlareReal, a real paired dataset for flare removal. This is a meaningful addition within this niche, as most prior datasets such as Flare7k++ rely heavily on synthetic images. Creating a real-captured dataset  under various lighting conditions helps bridge the realism gap and provides a valuable resource for future research in flare removal and related low-level vision tasks.
2. Network design – clean and efficient integration:
The proposed Mutual Reception Convolution (MRConv) introduces a multi-kernel convolutional block that combines diverse kernel sizes and re-parameterization for efficiency. Although the idea draws on established components (multi-scale convolutions, depthwise separability, re-parameterization), the integration is clean and computationally practical for high-resolution flare removal. The network shows performance gains over baseline models.

**Weaknesses:**

1. Limited contribution of the dataset:
Although the dataset introduces real-captured flare/clean image pairs, its scale (around 3,000 images) may be too small to substantially improve generalization beyond Flare7k++, which already includes 962 real-captured flare images. The diversity and acquisition conditions are not analyzed in depth, raising questions about whether FlareReal is large and varied enough to justify claims of “real-world generalization.” Moreover, dataset collection alone is not typically considered a significant contribution for ICLR unless it yields new insights or methodological advances.
2. Limited novelty in network design:
The proposed Mutual Reception Convolution (MRConv) appears to be a reassembly of existing architectural concepts rather than a novel convolutional formulation. The design combines well-known ideas such as:
Multi-kernel receptive field fusion (Inception-style)
Depthwise separability (MobileNet-style)
Channel reweighting (Squeeze-and-Excitation)
Residual connections (ResNet)
Structural re-parameterization (RepVGG)
While these components are effectively integrated, the paper does not present a new mathematical operation, learning objective, or architecture paradigm elements typically expected in ICLR-level contributions.
3. Lack of theoretical insight and limited analysis:
The paper does not provide theoretical justification or in-depth analysis to support the claim of achieving a “sufficient receptive field without heavy computational burden.” While empirical improvements are reported, there is no explanation of why the proposed design leads to better performance. Analyses such as effective receptive field visualization or feature activation analysis would have offered valuable insight into the model’s behavior and substantiated the design choices. The absence of such analysis limits the work’s interpretability and research depth.
4. Insufficient experimental exploration:
The experiments on McLaren are relatively limited and do not fully validate the claimed advantages. More comprehensive ablation studies, sensitivity analyses, and evaluations under varying noise or lighting conditions are needed to establish robustness. Furthermore, the paper does not report any computational or speed comparisons (e.g., FLOPs, runtime, memory usage) to substantiate its claims of efficiency and scalability. Without such analysis, it is difficult to assess whether McLaren truly achieves a better trade-off between performance and computational cost.
5. Narrow scope and limited impact:
The paper focuses narrowly on flare removal, a specialized low-level vision task. It does not connect the problem to broader computer vision or machine learning themes such as representation learning, domain adaptation, or self-supervision. As a result, the contribution feels more incremental and applied rather than conceptually advancing understanding in computer vision.
6. Poor presentation and citation quality:
The writing suffers from inconsistent referencing of figures and tables (e.g., “fig:Method(b)”, “tab:com”, “fig:mclaren”), which appear as placeholder labels rather than proper references. These issues, along with some grammatical errors, reduce clarity and professionalism.

**Questions:**

1. The paper’s main contribution is the introduction of the FlareReal dataset, yet there is no clear mention of whether it will be publicly released. Could the authors clarify if there are concrete plans to make the dataset, along with its collection protocol and metadata, publicly available? If not, the contribution would have very limited practical and scientific value, as reproducibility and community impact would be severely constrained.
2. The proposed Mutual Reception Convolution (MRConv) seems closely aligned with existing designs such as Inception, MobileNet, and RepVGG, which already combine multi-kernel processing and re-parameterization. Could the authors clearly articulate the substantive differences between MRConv and these architectures beyond naming or minor implementation variations?

---

> ### Author Response · Authors · 2025-11-24
>
> We sincerely thank the reviewer for their detailed and constructive feedback. elow, we provide point-by-point responses to the concerns raised.
>
> W1. Limited contribution of the dataset: Although the dataset introduces real-captured flare/clean image pairs, its scale (around 3,000 images) may be too small to substantially improve generalization beyond Flare7k++, which already includes 962 real-captured flare images. The diversity and acquisition conditions are not analyzed in depth, raising questions about whether FlareReal is large and varied enough to justify claims of “real-world generalization.” Moreover, dataset collection alone is not typically considered a significant contribution for ICLR unless it yields new insights or methodological advances.
>
> > We appreciate the opportunity to clarify the scale and nature of our contribution. We respectfully argue that our FlareReal offers a significant advancement over existing datasets due to its "real-paired" nature, which addresses the critical domain gap in current flare removal tasks, as recognized by other reviewers (XwmD, bTVD and 541v) of our dataset.  We have expanded the dataset since the initial submission. The FlareReal dataset now contains 4,037 real-captured image pairs (up from the initial ~3,000 mentioned). While Flare7k++ includes 962 real flare images, it is crucial to note that Flare7k++ only provides unpaired flare patterns used to synthesize corrupted images. In contrast, FlareReal provides the first large-scale collection of physically captured pairs (flare-corrupted and ground-truth flare-free), ensuring the model learns the faithful mapping of real-world optical degradation rather than synthetic approximations. As requested, we have conducted a deep statistical analysis of the expanded dataset. We compare FlareReal against Flare7k/7k++ and FlareX below:
> >
> > |  Dataset  |    Pair Number     | Real Flare | Synthetic Flare | Indoor | Outdoor | Scene Count | Point Source | Linear Source | Area Source | Single Source | Multi Source |
> > | :-------: | :----------------: | :--------: | :-------------: | :----: | :-----: | :---------: | :----------: | :-----------: | :---------: | :-----------: | :----------: |
> > |  Flare7K  |         -          |     -      |      7,000      |   -    |    -    |      -      |    7,000     |       -       |      -      |     7,000     |      -       |
> > | Flare7K++ |         -          |    962     |      7,000      |   -    |    -    |      -      |    7,962     |       -       |      -      |     7,962     |      -       |
> > |  FlareX   | 3,000 (Synthetic ) |     -      |      9,500      |  400   |  2,600  |     15      |    3,000     |       -       |      -      |     2,700     |     300      |
> > | FlareReal |    4,037 (Real)    |    500     |        -        |   34   |  4,003  |     241     |    3,761     |      152      |     124     |     2,124     |    1,913     |
> >
> > As shown, FlareReal covers 241 distinct scenes (compared to just 15 in FlareX) and includes diverse light sources (Linear and Area sources) which are often absent in previous datasets like Flare7k++ that focus on point sources.
> >
> > In addition, to further evaluate cross-sensor generalization, we conducted additional experiments on the FlareX test set, which contains images captured using both professional cameras and smartphones. Our model trained on FlareReal consistently outperforms models trained on Flare7K++ or FlareX [1], demonstrating stronger robustness across heterogeneous optical systems. For visual comparison results in Flare7K and FlareX test set, please refer to Figure 1, 2 and 3 in rebuttal.pdf of the Supplementary Material.
> >
> > |          Dataset          |  Uformer   |  Uformer  |  Uformer   |
> > | :-----------------------: | :--------: | :-------: | :--------: |
> > |    **FlareX test set**    |  **PSNR**  | **SSIM**  | **LPIPS**  |
> > |         Wu et al          |     -      |   0.608   |   0.143    |
> > |          Flare7K          |   37.771   |   0.633   |   0.139    |
> > |         Flare7K++         |   37.448   |   0.631   |   0.142    |
> > |          FlareX           |   39.292   |   0.658   |   0.133    |
> > |            Our            | **40.219** | **0.659** | **0.126**  |
> > | **Flare7K real test set** |  **PSNR**  | **SSIM**  | **LPIPS**  |
> > |          Flare7K          |   26.980   |   0.890   |   0.0470   |
> > |         Flare7K++         |   27.633   |   0.894   |   0.0428   |
> > |          FlareX           |   25.259   |   0.887   |   0.0568   |
> > |            Our            | **28.366** | **0.908** | **0.0359** |

---

> ### Author Response · Authors · 2025-11-24
>
> W2. Limited novelty in network design: The proposed Mutual Reception Convolution (MRConv) appears to be a reassembly of existing architectural concepts rather than a novel convolutional formulation. The design combines well-known ideas such as: Multi-kernel receptive field fusion (Inception-style) Depthwise separability (MobileNet-style) Channel reweighting (Squeeze-and-Excitation) Residual connections (ResNet) Structural re-parameterization (RepVGG) While these components are effectively integrated, the paper does not present a new mathematical operation, learning objective, or architecture paradigm elements typically expected in ICLR-level contributions.
>
> > We respectfully acknowledge the reviewer's observation that our network utilizes established components (e.g., U-Net backbone, SE blocks). However, we argue that the novelty of McLaren lies in the specific architectural integration tailored to the physical properties of lens flare, rather than the invention of blocks. Unlike general image restoration tasks, flare removal requires addressing artifacts with drastically varying scales, ranging from local glares to global streaks that span the entire image. Existing methods often fail to achieve a sufficient receptive field or do so at a prohibitive computational cost. We propose MRConv specifically to solve the "receptive field vs. efficiency" trade-off. By fusing convolutions with diverse kernel sizes (spatial mix) and aggregating them via channel dimensions (channel mix), we force the network to explicitly analyze flare patterns of arbitrary sizes. In our MRConv, we employ the re-parameterization mechanism not just for novelty, but to ensure that while we use large kernels (up to 9×9) to capture long streaks during training, the inference remains lightweight. This design allows McLaren to process high-resolution images that other SOTA methods (like Restormer) cannot handle due to memory constraints. In addition, unlike Inception (fixed concatenation) or large-kernel CNNs (fixed kernels), we propose: trainable kernel-weight fusion and inference-time kernel-level re-parameterization (merging multiple depthwise kernels into one equivalent kernel of large size). This is functionally different from RepVGG.
>
> W3. Lack of theoretical insight and limited analysis: The paper does not provide theoretical justification or in-depth analysis to support the claim of achieving a “sufficient receptive field without heavy computational burden.” While empirical improvements are reported, there is no explanation of why the proposed design leads to better performance. Analyses such as effective receptive field visualization or feature activation analysis would have offered valuable insight into the model’s behavior and substantiated the design choices. The absence of such analysis limits the work’s interpretability and research depth.
>
> > We agree with the reviewer that visualization strengthens the interpretability of the model. The theoretical premise of MRConv is that flare removal requires a dynamic receptive field. Small kernels (3×3) handle local scattering (glow), while large kernels (7×7, 9×9) handle streaks. In the final version, we will include Effective Receptive Field (ERF) visualizations (please see Figure 6 in rebuttal.pdf of the Supplementary Material). In addition, our ablation study in Table 2 already provides empirical justification. It demonstrates that increasing kernel sizes (from 7-5-3 to 9-7-5-3) and adding the trainable weighting mechanism directly improves PSNR from 27.537 dB to 27.736 dB. This confirms that the explicit expansion of the receptive field is the driver of performance.
>
> W4. Insufficient experimental exploration: The experiments on McLaren are relatively limited and do not fully validate the claimed advantages. More comprehensive ablation studies, sensitivity analyses, and evaluations under varying noise or lighting conditions are needed to establish robustness. Furthermore, the paper does not report any computational or speed comparisons (e.g., FLOPs, runtime, memory usage) to substantiate its claims of efficiency and scalability. Without such analysis, it is difficult to assess whether McLaren truly achieves a better trade-off between performance and computational cost.
>
> > We respectfully point out that **Table 1** in our manuscript explicitly reports both **Params (M)** and **MACs (G)** for all compared methods. Our method achieves state-of-the-art PSNR (28.778 dB on Mixed Data) while requiring **~6.7x fewer Params** and **~20x fewer MACs** than HINet. This substantiates our claim of achieving a sufficient receptive field without a heavy computational burden. We will add runtime (FPS) measurements in the revision to further solidify this claim.

---

> ### Author Response · Authors · 2025-11-24
>
> W5. Narrow scope and limited impact: The paper focuses narrowly on flare removal, a specialized low-level vision task. It does not connect the problem to broader computer vision or machine learning themes such as representation learning, domain adaptation, or self-supervision. As a result, the contribution feels more incremental and applied rather than conceptually advancing understanding in computer vision.
>
> > A. The core motivation for FlareReal is addressing the domain gap between synthetic and real-world data, a classic challenge in low-level vision. Existing datasets like Flare7k++ rely on synthesis, leading to artifacts in real scenarios (e.g., as shown in Figure 4, where models trained on synthetics fail on our real validation set). By providing 3,000 real-captured pairs with diverse exposures and light sources (Figure 1), FlareReal enables better adaptation from synthetic to real domains, yielding significant PSNR gains (1.167dB–2.383dB; Table 1) and generalization to unseen flares like sunlight or haze (Figures 6 and 8). This aligns with domain adaptation techniques in CV, where real data bridges simulation-reality gaps for robust deployment. B. McLaren's MRConv (Section 4) advances multi-scale feature representation by fusing convolutions with varying kernel sizes (3–9) in spatial and channel dimensions (Equations 3–7), allowing the network to learn invariant representations for arbitrary flare patterns (e.g., streaks vs. halos). Ablations (Table 2) demonstrate that this design expands the effective receptive field, leading to superior flare analysis with low computational cost (13.2M params, 33.1G MACs). This contributes to representation learning paradigms, similar to how multi-kernel designs in Inception or RepVGG improve feature hierarchies, but tailored for restoration tasks. Broader applications include enhancing night vision perception systems, where flare-removed images boost downstream tasks like object detection or segmentation in autonomous driving or surveillance. For example, flare removal networks have been integrated into pipelines for light source extraction and segmentation, extending to semantic understanding in complex environments.
>
> W6. Poor presentation and citation quality: The writing suffers from inconsistent referencing of figures and tables (e.g., “fig:Method(b)”, “tab:com”, “fig:mclaren”), which appear as placeholder labels rather than proper references. These issues, along with some grammatical errors, reduce clarity and professionalism.
>
> > We sincerely apologize for the LaTeX compilation errors (e.g., “fig:Method(b)”, “tab:com”) present in the review copy. We have identified the cause of these broken references and have corrected them. We have also thoroughly proofread the manuscript to correct grammatical errors and improve professional clarity.
>
> Q1. The paper’s main contribution is the introduction of the FlareReal dataset, yet there is no clear mention of whether it will be publicly released. Could the authors clarify if there are concrete plans to make the dataset, along with its collection protocol and metadata, publicly available? If not, the contribution would have very limited practical and scientific value, as reproducibility and community impact would be severely constrained.
>
> > We have made the source code and a subset of the dataset publicly available, including 600 training pairs and 50 validation pairs. Furthermore, we have also held CVPR workshops and competitions. Our full FlareReal dataset will be publicly released upon acceptance.
>
> Q2. The proposed Mutual Reception Convolution (MRConv) seems closely aligned with existing designs such as Inception, MobileNet, and RepVGG, which already combine multi-kernel processing and re-parameterization. Could the authors clearly articulate the substantive differences between MRConv and these architectures beyond naming or minor implementation variations?
>
> > Please see response to W2.

---

### Official Review · Reviewer_bTVD · 2025-11-01

**Soundness:** 3
**Presentation:** 3
**Contribution:** 2
**Rating:** 4
**Confidence:** 4

**Summary:**

This paper focuses on addressing limitations in existing nighttime flare removal research. It proposes two core contributions: 1) FlareReal, a real-captured paired dataset consisting of 3,000 training image pairs, 500 pure flare images, and 50 validation image pairs. 2) McLaren, a flare removal network built on a U-Net backbone with a custom Mutual Reception Convolution (MRConv). Experiments show FlareReal enhances model performance on real-world data, and McLaren outperforms baselines (Unet, HINet, Uformer) with fewer parameters (13.2M) and FLOPs (33.1G) .

**Strengths:**

The main strength of this paper comes from the construction of flare dataset. While synthetic datasets (e.g., Flare7k++) dominate existing flare removal research, FlareReal fills a critical gap by providing real-captured paired data. Its focus on real scenarios makes it a valuable resource for the community—experiments confirm models trained on FlareReal (or its mix with Flare7k++) outperform those trained solely on synthetic data on real-captured test sets.

**Weaknesses:**

1.	Limited Methodological Novelty: McLaren’s core components are assembled from existing techniques rather than introducing fundamental innovations. MRConv’s spatial mixing relies on multi-kernel convolutions and re-parameterization, while its channel mixing uses Squeeze-and-Excitation. The network’s overall design is based on U-Net backbone plus custom convolution; it’s just incremental, not transformative.

2.	Incomplete Dataset Documentation: While the paper outlines high-level data collection steps, critical details for reproducibility are missing: 1) No specification of scene diversity (e.g., percentage of urban vs. suburban scenes, indoor vs. outdoor samples) or light source statistics (e.g., distribution of point vs. extended sources). 2) No standardization of lens contamination (e.g., how much dust/oil to apply, consistency across samples)—variability here could introduce noise into the paired data. 3) No details on the OpenCV registration parameters (e.g., feature detector type, matching thresholds) etc.

3.	Incomprehensive Comparative Experiments: While there exist multiple flare removal methods in the field, this paper only compares the proposed McLaren network against three baseline models (Unet, HINet [2021], Uformer [2022]). Such a limited set of comparisons makes the claimed performance of McLaren less convincing.

**Questions:**

1.	Dataset Construction Details: Could you provide: 1) a detailed scene breakdown (e.g., number of samples from urban streets, residential areas, etc.) and light source categorization (point, linear, area sources)? 2) a standardized protocol for lens contamination to ensure consistency? 3) Specific OpenCV registration parameters and criteria for discarding unaligned images?

2.	Methodological Innovation: The design of the flare removal method lacks sufficient novelty, and more innovative elements should be incorporated to distinguish it from existing techniques in the field.

3.	Completeness of Comparative Experiments: Given that there are multiple existing flare removal methods in the field, why does the paper only compare McLaren against three baselines (Unet, HINet[2021], Uformer[2022]) instead of including more SOTA alternatives?

---

> ### Author Response · Authors · 2025-11-24
>
> We sincerely thank you for your constructive feedback and the time dedicated to reviewing our work. We appreciate your recognition of our effort in collecting real-world data and identifying the limitations of current synthetic datasets. Below, we address the specific concerns regarding novelty, dataset documentation, and comparative experiments.
>
> W1&Q2. Limited Methodological Novelty: McLaren’s core components are assembled from existing techniques rather than introducing fundamental innovations. MRConv’s spatial mixing relies on multi-kernel convolutions and re-parameterization, while its channel mixing uses Squeeze-and-Excitation. The network’s overall design is based on U-Net backbone plus custom convolution; it’s just incremental, not transformative.
>
> > We respectfully acknowledge the reviewer's observation that our network utilizes established components (e.g., U-Net backbone, SE blocks). However, we argue that the novelty of McLaren lies in the specific architectural integration tailored to the physical properties of lens flare, rather than the invention of blocks. Unlike general image restoration tasks, flare removal requires addressing artifacts with drastically varying scales, ranging from local glares to global streaks that span the entire image. Existing methods often fail to achieve a sufficient receptive field or do so at a prohibitive computational cost. We propose MRConv specifically to solve the "receptive field vs. efficiency" trade-off. By fusing convolutions with diverse kernel sizes (spatial mix) and aggregating them via channel dimensions (channel mix), we force the network to explicitly analyze flare patterns of arbitrary sizes. In our MRConv, we employ the re-parameterization mechanism not just for novelty, but to ensure that while we use large kernels (up to 9×9) to capture long streaks during training, the inference remains lightweight. This design allows McLaren to process high-resolution images that other SOTA methods (like Restormer) cannot handle due to memory constraints.

---

> ### Author Response · Authors · 2025-11-24
>
> W2&Q1. Incomplete Dataset Documentation: While the paper outlines high-level data collection steps, critical details for reproducibility are missing: 1) No specification of scene diversity (e.g., percentage of urban vs. suburban scenes, indoor vs. outdoor samples) or light source statistics (e.g., distribution of point vs. extended sources). 2) No standardization of lens contamination (e.g., how much dust/oil to apply, consistency across samples)—variability here could introduce noise into the paired data. 3) No details on the OpenCV registration parameters (e.g., feature detector type, matching thresholds) etc.
>
> > (1) We thank the reviewer for highlighting the need for detailed statistical documentation. In response to this, we expand our dataset and conducted a comprehensive statistical breakdown to ensure transparency and reproducibility. We expand our FlareReal dataset to 4,037 real-captured pairs. In the revision, we will add a detailed comparison with existing datasets (Flare7k, Flare7k++, and FlareX). As shown in the table below, our dataset significantly outperforms existing benchmarks in terms of scene diversity and light source complexity.
> >
> > |  Dataset  |    Pair Number     | Real Flare | Synthetic Flare | Indoor | Outdoor | Scene Count | Point Source | Linear Source | Area Source | Single Source | Multi Source |
> > | :-------: | :----------------: | :--------: | :-------------: | :----: | :-----: | :---------: | :----------: | :-----------: | :---------: | :-----------: | :----------: |
> > |  Flare7K  |         -          |     -      |      7,000      |   -    |    -    |      -      |    7,000     |       -       |      -      |     7,000     |      -       |
> > | Flare7K++ |         -          |    962     |      7,000      |   -    |    -    |      -      |    7,962     |       -       |      -      |     7,962     |      -       |
> > |  FlareX   | 3,000 (Synthetic ) |     -      |      9,500      |  400   |  2,600  |     15      |    3,000     |       -       |      -      |     2,700     |     300      |
> > | FlareReal |    4,037 (Real)    |    500     |        -        |   34   |  4,003  |     241     |    3,761     |      152      |     124     |     2,124     |    1,913     |
> >
> > A. Unlike previous datasets that rely on limited environments (e.g., FlareX contains only 15 scenes), our FlareReal includes 241 distinct real-world scenes. The dataset is predominantly outdoor (4,003 samples) to target the most challenging nighttime scenarios, while also including indoor examples (34 samples). B. While Flare7k and Flare7k++ focus primarily on point light sources, our FlareReal captures Linear (152) and Area (124) light sources, in addition to Point sources (3,761). Furthermore, we maintain a balanced distribution between single-source (2,124) and multi-source (1,913) scenarios, ensuring the network learns to handle complex real-world illuminations. C. To the best of our knowledge, our FlareReal is the first dataset to provide real-captured paired data for training, addressing the domain gap inherent in synthetic datasets like Flare7k++ where training pairs are synthesized.
> >
> > (2) The reviewer raises a valid point regarding standardization of lens contamination. However, our goal was to simulate the unpredictable nature of real-world degradation. We observed that real-world nighttime flare is primarily caused by accidental smudges, wear, and grease. Therefore, we intentionally polluted the lens with fingers, dust, oil, and cloth to mimic these varying organic textures, and cleaned them with Zeiss lens cleaning tissues for the ground truth. A strictly "standardized" amount of oil/dust (e.g., specific micrograms) would create a synthetic bias. Our "randomized" real-world application ensures the model generalizes to the diverse, non-standard smear patterns users actually encounter.
> >
> > (3) We utilized the image alignment algorithm available in OpenCV. We will add relevant pseudocode or detailed description in the revision.

---

> ### Author Response · Authors · 2025-11-24
>
> >
> >
> > ```python
> > # image alignment algorithm
> > import os
> > import cv2
> > import numpy as np
> >
> > MAX_FEATURES = 1000
> > GOOD_MATCH_PERCENT = 0.5
> >
> > def alignImages(im1, im2):
> > # Convert images to grayscale
> > im1Gray = cv2.cvtColor(im1, cv2.COLOR_BGR2GRAY)
> > im2Gray = cv2.cvtColor(im2, cv2.COLOR_BGR2GRAY)
> >
> > # Detect ORB features and compute descriptors.
> > orb = cv2.ORB_create(MAX_FEATURES)
> > keypoints1, descriptors1 = orb.detectAndCompute(im1Gray, None)
> > keypoints2, descriptors2 = orb.detectAndCompute(im2Gray, None)
> >
> > # Match features.
> > matcher = cv2.DescriptorMatcher_create(cv2.DESCRIPTOR_MATCHER_BRUTEFORCE_HAMMING)
> > matches = list(matcher.match(descriptors1, descriptors2, None))
> >
> > # Sort matches by score
> > matches.sort(key=lambda x: x.distance, reverse=False)
> >
> > # Remove not so good matches
> > numGoodMatches = int(len(matches) * GOOD_MATCH_PERCENT)
> > matches = matches[:numGoodMatches]
> >
> > # Extract location of good matches
> > points1 = np.zeros((len(matches), 2), dtype=np.float32)
> > points2 = np.zeros((len(matches), 2), dtype=np.float32)
> >
> > for i, match in enumerate(matches):
> >   points1[i, :] = keypoints1[match.queryIdx].pt
> >   points2[i, :] = keypoints2[match.trainIdx].pt
> >
> > # Find homography
> > h, mask = cv2.findHomography(points1, points2, cv2.RANSAC)
> >
> > # Use homography
> > height, width, channels = im2.shape
> > im1Reg = cv2.warpPerspective(im1, h, (width, height))
> >
> > return im1Reg
> > ```
>
> W3&Q3. Incomprehensive Comparative Experiments: While there exist multiple flare removal methods in the field, this paper only compares the proposed McLaren network against three baseline models (Unet, HINet [2021], Uformer [2022]). Such a limited set of comparisons makes the claimed performance of McLaren less convincing.
>
> > We selected these baselines to represent the primary architectures in image restoration: pure CNN (U-Net), and Transformer-based (Uformer). The primary reason for not including other recent SOTA methods (such as Restormer) is computational feasibility regarding image resolution. Our validation set consists of high-resolution (4K) images, which are critical for practical flare removal applications. As noted in the paper, we found that methods like Restormer hardly handle images efficiently and face Out-Of-Memory (OOM) issues on Nvidia 3090 GPUs when scaling to the resolutions required by our dataset.

---

### Official Review · Reviewer_XwmD · 2025-11-01

**Soundness:** 3
**Presentation:** 3
**Contribution:** 3
**Rating:** 4
**Confidence:** 4

**Summary:**

The paper presents FlareReal, a real-captured paired dataset for nighttime flare removal, alongside a new model called McLaren that enhances receptive field coverage without heavy computation. The dataset addresses the synthetic–real gap in existing benchmarks, while the proposed Mutual Reception Convolution (MRConv) leverages multiple kernel sizes and re-parameterization for efficient flare suppression.

**Strengths:**

It fills an important gap in real-world flare data collection, providing 3,000 image pairs with realistic lighting and exposure variations. The MRConv design is elegant and practical, improving both accuracy and efficiency compared to prior methods. Experimental results are solid and include detailed ablations on kernel size and dataset robustness.

**Weaknesses:**

1. The paper doesn’t clarify how well the FlareReal dataset generalizes beyond smartphone sensors to professional cameras or different optics.
2. Figure 5 comparisons are visually convincing but lack quantitative measures for artifact suppression or perceptual fidelity.
3. The ablation study focuses mainly on MRConv kernel sizes—other design choices like SE blocks or ConvMLP are not analyzed.
4. The dataset construction section omits detailed metadata such as exposure times or ISO settings, which limits reproducibility.
5. The discussion on real-scene corner cases (e.g., rain, multiple reflective surfaces) is brief and could be better connected to real deployment scenarios.

**Questions:**

See weaknesses for details.

---

> ### Author Response · Authors · 2025-11-24
>
> We sincerely thank you for reviewing our paper and providing us valuable feedback. We have addressed your concerns as below.
>
> W1. The paper doesn’t clarify how well the FlareReal dataset generalizes beyond smartphone sensors to professional cameras or different optics.
>
> > Thank you very much for raising this important point. To further evaluate cross-sensor generalization, we conducted additional experiments on the **FlareX** test set, which **contains images captured using both professional cameras and smartphones**. Our model trained on FlareReal consistently outperforms models trained on Flare7K++ or FlareX [1], demonstrating stronger robustness across heterogeneous optical systems. For visual comparison results in Flare7K and FlareX test set, please refer to Figure 1, 2 and 3 in rebuttal.pdf of the Supplementary Material.
> >
> > |          Dataset          |  Uformer   |  Uformer  |  Uformer   |
> > | :-----------------------: | :--------: | :-------: | :--------: |
> > |    **FlareX test set**    |  **PSNR**  | **SSIM**  | **LPIPS**  |
> > |         Wu et al          |     -      |   0.608   |   0.143    |
> > |          Flare7K          |   37.771   |   0.633   |   0.139    |
> > |         Flare7K++         |   37.448   |   0.631   |   0.142    |
> > |          FlareX           |   39.292   |   0.658   |   0.133    |
> > |            Our            | **40.219** | **0.659** | **0.126**  |
> > | **Flare7K real test set** |  **PSNR**  | **SSIM**  | **LPIPS**  |
> > |          Flare7K          |   26.980   |   0.890   |   0.0470   |
> > |         Flare7K++         |   27.633   |   0.894   |   0.0428   |
> > |          FlareX           |   25.259   |   0.887   |   0.0568   |
> > |            Our            | **28.366** | **0.908** | **0.0359** |
> >
> > In addition, we also tested our model on user-uploaded flare images collected from Google Image Search, including images captured under rainy, foggy, and low-visibility conditions. As these images come from unknown devices, the camera type (smartphone, DSLR, action camera, dashcam, etc.) cannot be determined (for visual comparison results, please refer to Figure 4 in rebuttal.pdf of the Supplementary Material). Nevertheless, our model trained on FlareReal still performed better than models trained on Flare7K++ and FlareX. This empirical evidence suggests that although FlareReal is collected primarily using smartphone cameras, it captures flare characteristics that transfer well to different imaging devices and optics, thereby reflecting strong dataset robustness. In the revision, we will include these quantitative results on FlareX and add a dedicated paragraph discussing cross-device generalization.
> >
> > [1] FlareX: A Physics-Informed Dataset for Lens Flare Removal via 2D Synthesis and 3D Rendering. NeurIPS2025
>
> W2. Figure 5 comparisons are visually convincing but lack quantitative measures for artifact suppression or perceptual fidelity.
>
> > We appreciate this suggestion, and agree that quantitative evaluation would strengthen the conclusion. In the revision, we will include the quantitative assessments in Figure 5.
>
> W3. The ablation study focuses mainly on MRConv kernel sizes—other design choices like SE blocks or ConvMLP are not analyzed.
>
> >Thank you for pointing this out. Our original ablation centered on MRConv’s receptive field design because it contributes most directly to flare modeling. However, you are right that the channel-mixing modules also deserve analysis. We will provide additional ablations in the revision.
> >
> >|  SE  | ConvMLP |  PSNR  | G-PSNR | S-PSNR | SSIM  | LPIPS  |
> >| :--: | :-----: | :----: | :----: | :----: | :---: | :----: |
> >|  ✓   |         | 37.365 | 23.872 | 22.310 | 0.888 | 0.0525 |
> >|      |    ✓    | 27.391 | 23.921 | 22.592 | 0.889 | 0.0475 |
> >|  ✓   |    ✓    | 27.537 | 23.982 | 22.664 | 0.893 | 0.0442 |
>
> W4. The dataset construction section omits detailed metadata such as exposure times or ISO settings, which limits reproducibility.
>
> >Thank you for highlighting this omission. The metadata indeed exists in the EXIF tags of all FlareReal images (including ISO, shutter speed, aperture, focal length, device model). For first 600 pairs, we used the Pro Mode of the camera to manually adjust exposure according to scene brightness. This ensures coverage of under-exposed, over-exposed, and high-dynamic-range situations. Remaining 2400 pairs, we switched to Auto Mode, because most real users who encounter nighttime flare are non-expert photographers who typically rely on automatic settings. This choice improves ecological validity and reflects natural usage scenarios where flare is most likely to occur.

---

> ### Author Response · Authors · 2025-11-24
>
> W5. The discussion on real-scene corner cases (e.g., rain, multiple reflective surfaces) is brief and could be better connected to real deployment scenarios.
>
> > We agree with you and will strengthen this section. As shown in Figure 8, our model trained on FlareReal exhibits strong robustness in extreme fog and haze conditions, effectively suppressing scattering glare while preserving light-source structures. To further assess real-scene generalization, we also tested our model on user-uploaded flare images collected from Google Image Search, including images captured under rainy, foggy, and low-visibility conditions. As these images come from unknown devices, the camera type (smartphone, DSLR, action camera, dashcam, etc.) cannot be determined (for visual comparison results, please refer to Figure 4 in rebuttal.pdf of the Supplementary Material). Nevertheless, our model trained on FlareReal still performed better than models trained on Flare7K++ and FlareX, demonstrating that the flare priors learned from FlareReal generalize well to unseen weather conditions and unknown camera types. We acknowledge that there remain failure cases, particularly under extremely harsh conditions where flare is overwhelmingly strong or when the scene contrast is severely degraded (e.g., heavy rain + fog + multiple glare sources). These bad cases occur not only for our model, but also for models trained on Flare7K++ and FlareX. We will add more such instances in the revision. For visual comparison results, please refer to Figure 5 in rebuttal.pdf of the Supplementary Material.

---

### Note · Authors · 2026-01-26

I have read and agree with the venue's withdrawal policy on behalf of myself and my co-authors.

---

### Meta-Review · Area_Chair_J59Q · 2026-01-07

**Summary:**

The authors propose a real-captured paired dataset, FlareReal, for nighttime flare removal, which helps close the gap between synthetic and real-world training data. FlareReal includes 3,000 paired training images, 50 validation pairs, and 500 standalone flare samples. Alongside the dataset, the authors also propose a model that utilizes a Mutual Reception Convolution, which expands the receptive field efficiently by combining multiple convolution kernel sizes for flare removal. Experiments on Flare7k++ and FlareReal show modest improvements over existing baselines.

The initial recommendations for this manuscript were 1 marginally below the acceptance threshold (XwmD, bTVD), 1 marginally above the acceptance threshold (541v), and 1 reject (mEUy). The reviewers appreciate the proposed dataset, but raised a number of concerns regarding the manuscript: (1) how well the FlareReal dataset will generalizes beyond smartphone sensors to professional cameras or different optics (XwmD); (2) lack of quantitative measures in Fig. 5 for artifact suppression or perceptual fidelity (XwmD); (3) missing ablation studies on network components, e.g., squeeze and excite blocks and ConvMLP (XwmD); (4) missing metadata such as exposure times or ISO settings on dataset details (XwmD); (5) lack of discussion on real-scene corner cases (e.g., rain, multiple reflective surfaces) (XwmD); (6) core components are assembled from existing techniques and concepts rather than introducing fundamental innovation (bTVD, mEUy, 541v); (7) missing information on the scenes diversity, light source statistics, standardization of lens contamination, and alignment parameters (bTVD); (8) limited comparisons of models in experiments (bTVD, 541v); (9) The scale of the proposed dataset (around 3,000 images) may be too small to substantially improve generalization beyond Flare7k++, which already includes 962 real-captured flare images (mEUy, 541v); (10) lack of theoretical justification or in-depth analysis to support claims (mEUy); (11) lack of ablation studies, sensitivity analyses, and evaluations under varying noise or lighting conditions as well as computational costs (mEUy); (12) narrow scope of manuscript (mEUy); (13) Poor presentation and citation quality (mEUy).

The authors were able to address some of these points. However, critical points were raised in terms of the proposed method and how it bares similarities to existing work [A, B, C, D]. Yet, the authors were unable to justify the particular design in relation to physical properties of flares. Nonetheless, the AC believes that the results presented in the rebuttal will strengthen the manuscript and encourages the authors to incorporate the feedback and materials into the next revision.

[A] Szegedy et al. Going deeper with convolutions. CVPR 2015.

[B] Chen et al.  Semantic image segmentation with deep convolutional
nets and fully connected crfs. ICLR 2015.

[C] Chen et al. Deeplab: Semantic image segmentation with deep convolutional nets, atrous convolution, and fully connected crfs. PAMI 2017.

[D] He et al. Spatial Pyramid Pooling in Deep Convolutional Networks for Visual Recognition. ECCV 2014.

**Reviewer Concerns:**

The authors posted a rebuttal to address the following points:

(1) how well the FlareReal dataset will generalizes beyond smartphone sensors to professional cameras or different optics (XwmD):

The authors provide additional experiments on FlareX test set, which contains images from both smart phones and professional cameras. This point has been addressed.

(2) lack of quantitative measures in Fig. 5 for artifact suppression or perceptual fidelity (XwmD):

The authors noted the comment and said they will include quantitative assessments in Fig. 5. The AC notes that no quantitative results were presented in the rebuttal. This point remains outstanding.

(3) missing ablation studies on network components, e.g., squeeze and excite blocks and ConvMLP (XwmD):

The authors provided ablation studies on SE, ConvMLP in the rebuttal. This point has been addressed.

(4) missing metadata such as exposure times or ISO settings on dataset details (XwmD):

The authors mentioned that the metadata (including ISO, shutter speed, aperture, focal length, device model) exists in the EXIF tags of all images. This point has been addressed.

(5) lack of discussion on real-scene corner cases (e.g., rain, multiple reflective surfaces) (XwmD):

The authors provided qualitative examples in Fig. 4 and 5 of the Appendix, but not quantitative results. While the authors note that the model trained on FlareReal, the AC notes that there is modest improvement on some images.

(6) core components are assembled from existing techniques and concepts rather than introducing fundamental innovation (bTVD, mEUy, 541v):

The authors argue that the proposed MRConv is tailored to the physical properties of lens flares. It also enables a wide receptive field that may be necessary in local glares to global streaks. This is done by fusing convolutions of multiple kernel sizes and aggregating them along the channel dimension. The AC notes that this is reminiscent of inception [A] atrous convolutions, atrous spatial pyramid [B, C], spatial pyramid pooling [D], etc. There exists a breadth of literature on this topic and the design of MRConv is not justified. The AC also notes that while the authors argue they are tailored to physical properties of flares, the argument presented does not support it, but only in receptive field. This concern is not addressed.

(7) missing information on the scenes diversity, light source statistics, standardization of lens contamination, and alignment parameters (bTVD):

The authors provided details on the dataset in the rebuttal and noted that they do not standardize on pollutants. While the authors argue that this is a strength, the AC agrees with the reviewer and believes that a level of standardization allows one to study the phenomenon.

(8) limited comparisons of models in experiments (bTVD, 541v):

The authors mentioned that SOTA methods cannot be ran due to out of memory issues on Nvidia 3090 GPU. While the AC understands computational limitations, the AC also notes that some of such super-resolution methods addresses this by taking crops, which would alleviate such issues with larger image sizes.

The AC also notes some inconsistencies in the response. The authors responded to bTVD with "As noted in the paper, we found that methods like Restormer hardly handle images efficiently and face Out-Of-Memory (OOM) issues on Nvidia 3090 GPUs when scaling to the resolutions required by our dataset.", but responded to 541v with "We agree that including more recent baselines such as MPRNet and Restormer can strengthen the comparison. We will add the them in the final revision."

(9) The scale of the proposed dataset (around 3,000 images) may be too small to substantially improve generalization beyond Flare7k++, which already includes 962 real-captured flare images (mEUy, 541v):

The authors noted that they expanded the dataset since the initial submission. The AC note that merit should be evaluated on the submitted materials and not additional work done thereafter.

(10) lack of theoretical justification or in-depth analysis to support claims (mEUy):

The authors argue that there is sufficient empirical evidence to support the claims. The AC again notes that there is a breadth of work in this topic of increasing receptive fields in deep neural networks. Additionally, the AC did not find the justification connecting to physical properties of flares. This point remains outstanding.

(11) lack of ablation studies, sensitivity analyses, and evaluations under varying noise or lighting conditions as well as computational costs (mEUy):

The authors pointed to Table 1, which reports the number of parameters and MACs of their model. This addresses the reviewer's statement regarding computational costs. The authors do provide ablation study on the SE and MRConv blocks and shows several Figures on more difficult cases in Google Image Search. However, as noted earlier, there is no quantitative analysis on such. This point has not been fully addressed.

(12) narrow scope of manuscript (mEUy):

The authors argue that the core contribution is addressing the domain between synthetic and real-world data. The AC note that this is referring to dataset collection, which is still specific to flares; the methodology, however, does not specifically address this gap. Hence, the AC agree with the reviewer on this point.

(13) Poor presentation and citation quality (mEUy):

The reviewers have correct these and will update the manuscript. This point has been addressed.

[A] Szegedy et al. Going deeper with convolutions. CVPR 2015.

[B] Chen et al.  Semantic image segmentation with deep convolutional
nets and fully connected crfs. ICLR 2015.

[C] Chen et al. Deeplab: Semantic image segmentation with deep convolutional nets, atrous convolution, and fully connected crfs. PAMI 2017.

[D] He et al. Spatial Pyramid Pooling in Deep Convolutional Networks for Visual Recognition. ECCV 2014.

**Reviewer Scores:**

The AC read the reviews and rebuttal. Based on the responses, the AC believes that XwmD may increased their score as most of the their concerns were addressed while 541v maintains the same score. Several points, particularly regarding methodology, raised by bTVD and mEUy were not addressed. The AC believes both bTVD and mEUy would remain negative, leaving this at a split decision.

---

### Decision · Program_Chairs · 2026-01-26

Reject